ecology

biodiversity, ecosystem functioning, environmental gradients, resource supply, chemosynthesis, methane seep

**Author for correspondence:**
Oliver S. Ashford
e-mail: oashford@ucsd.edu

# Relationships between biodiversity and ecosystem functioning proxies strengthen when approaching chemosynthetic deep-sea methane seeps

Oliver S. Ashford[1], Shuzhe Guan[1], Dante Capone[1,2], Katherine Rigney[1,3], Katelynn Rowley[1], Erik E. Cordes[4], Jorge Cortés[5], Greg W. Rouse[1], Guillermo F. Mendoza[1], Andrew K. Sweetman[6] and Lisa A. Levin[1,7]

[1]Integrative Oceanography Division, Scripps Institution of Oceanography, University of California San Diego, San Diego, CA 92007, USA
[2]University of California, Santa Cruz, CA 95064, USA
[3]Carleton College, Northfield, MN 55057, USA
[4]Department of Biology, Temple University, Temple, PA 19122, USA
[5]CIMAR, Universidad de Costa Rica, San José, Costa Rica
[6]The Lyell Centre for Earth and Marine Science and Technology, Heriot-Watt University, Edinburgh, UK
[7]Center for Marine Biodiversity and Conservation, Scripps Institution of Oceanography, University of California San Diego, San Diego, CA 92093, USA

OSA, 0000-0001-5473-7057; EEC, 0000-0002-6989-2348; JC, 0000-0001-7004-8649; AKS, 0000-0002-9547-9493; LAL, 0000-0002-2858-8622

As biodiversity loss accelerates globally, understanding environmental influence over biodiversity–ecosystem functioning (BEF) relationships becomes crucial for ecosystem management. Theory suggests that resource supply affects the shape of BEF relationships, but this awaits detailed investigation in marine ecosystems. Here, we use deep-sea chemosynthetic methane seeps and surrounding sediments as natural laboratories in which to contrast relationships between BEF proxies along with a gradient of trophic resource availability (higher resource methane seep, to lower resource photosynthetically fuelled deep-sea habitats). We determined sediment fauna taxonomic and functional trait biodiversity, and quantified bioturbation potential (BPc), calcification degree, standing stock and density as ecosystem functioning proxies. Relationships were strongly unimodal in chemosynthetic seep habitats, but were undetectable in transitional 'chemotone' habitats and photosynthetically dependent deep-sea habitats. In seep habitats, ecosystem functioning proxies peaked below maximum biodiversity, perhaps suggesting that a small number of specialized species are important in shaping this relationship. This suggests that absolute biodiversity is not a good metric of ecosystem 'value' at methane seeps, and that these deep-sea environments may require special management to maintain ecosystem functioning under human disturbance. We promote further investigation of BEF relationships in non-traditional resource environments and emphasize that deep-sea conservation should consider 'functioning hotspots' alongside biodiversity hotspots.

## 1. Introduction

The ecosystem services associated with biologically diverse, efficiently functioning ecosystems are numerous, economically valuable and fundamental to human wellbeing worldwide [1–4]. For example, high-functioning ecosystems supply humanity with clean water, food, medicine and energy, regulating the composition of the air we breathe and even inspiring art and other activities of cultural significance, among other contributions [1,4–6]. However, natural

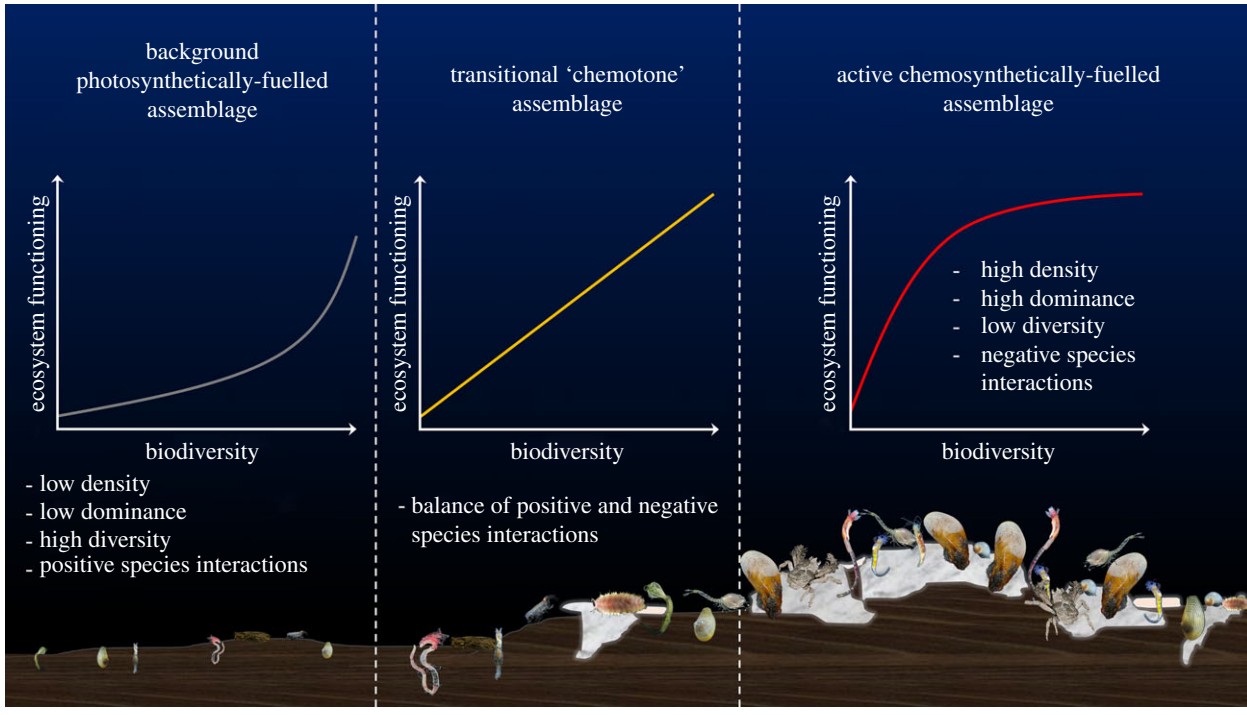

**Figure 1.** Hypotheses for the form of BEF relationships in contrasting deep-sea environments; lower resource photosynthetically dependent background habitat: grey line, accelerating form; transitional 'chemotone' habitat: orange line, linear form; higher resource chemosynthetically dependent methane seep habitat: red line, saturating form. (Online version in colour.)

environments are being degraded as a result of human actions, resulting in a high rate of biodiversity loss worldwide [7,8].

With growing recognition of this, since the mid-1990s scientists have been working to understand the relationship between biodiversity and the functioning of ecosystems [9–12]. Early experimental studies were generally small in scale, limited in complexity, highly controlled and focussed only on temperate terrestrial systems, but demonstrated a positive link between biodiversity–ecosystem functioning (BEF) [10,12–14]. Positive BEF relationships have since been reported for multiple taxa at various trophic levels, in a selection of environments, in both experimental and natural settings, and for a variety of ecosystem functioning proxies [15–19]. However, very few of these studies have investigated BEF relationships in the deep ocean and no studies have investigated these relationships in chemosynthetic environments.

The shape of BEF relationships has been shown to vary even within photosynthesis-based ecosystems and taxonomic groups [20]. For example, positive BEF relationships in deep-sea nematode assemblages have variously been reported to be linear, asymptotic (saturating) or exponential (accelerating) in form [18,21–23]. Understanding the drivers of this variability is key to determining how rapidly ecosystem functioning may be lost with declining biodiversity as a result of natural or anthropogenic changes (figure 1; electronic supplementary material, figure S1), with important implications for the effective management of natural environments

The prevailing environmental conditions experienced by assemblages may be a driver of the form of BEF relationships through control over species occurrence, density, behaviour and physiology, and hence the types and rates of biological processes occurring, and the outcomes of interactions among species [11,20,24–27] (electronic supplementary material, table S1). Environmental conditions may thus control the identity of functional traits present in an assemblage, their relative dominance and the ways in

which these traits interact, influencing ecosystem functioning [12,28].

Resource supply gradients play an important role in controlling assemblage diversity [29–31] and are often associated with changes in abiotic environmental conditions, but relatively few studies have considered the influence of this factor over the form of BEF relationship [30,32–34], and none have investigated this in chemosynthetic ecosystems such as methane seeps. Resource supply may influence the form of BEF relationships through its control over organismal diversity, dominance, density and composition, and, assuming a constant degree of disturbance, the competitive landscape experienced by assemblages [29,30,35,36]. When resource supply is elevated, organismal densities are expected to be relatively high, promoting negative species interactions, such as competition, and high assemblage dominance. In deep-sea environments, this has been hypothesized to produce BEF relationships which saturate quickly with increasing diversity [20]. By contrast, when resource supply is relatively low, organismal densities and assemblage dominance are expected to be lower, promoting positive species interactions, such as facilitation, and high assemblage diversity. In deep-sea environments, this has been hypothesized to produce BEF relationships which become increasingly positive with increasing diversity [18,20]. At intermediate levels of resource supply, positive and negative species interactions may balance each other, producing linear BEF relationships [20] (figure 1). Negative BEF relationships are also possible [37,38] and may occur when increasing diversity promotes increasing competition from species with relatively low contributions to ecosystem functioning, reducing the abundances of species with relatively high contributions to ecosystem functioning [39].

Greater insight into the influence of environmental context on the form of relationship between BEF could be gained by exploring natural gradients [27,40]. Here, we undertake a mensurative study, using deep-sea methane seeps and the

surrounding seafloor as a natural laboratory in which to investigate how the form of relationship between BEF proxies changes along gradients of resource supply; from lower resource deep-sea habitats primarily dependent upon the delivery of photosynthetically fixed carbon from surface waters, to higher resource chemosynthetically fuelled seep habitats. Deep-sea methane seeps are resource hotspots in what is typically considered a food-limited environment [41–43]. Steep gradients of resource supply and origin surround methane seeps, as well as gradients in other environmental variables, including sediment and water chemistry (and hence physiological stress associated with reducing fluids low in oxygen and high in hydrogen sulfide concentrations), substratum type and turbidity [41].

We hypothesized that the shape of relationship between BEF proxies would change when moving from photosynthetically dependent deep-sea habitats towards chemosynthetically dependent methane seep habitats (figure 1). We predicted that relationships between BEF proxies at methane seeps (high-resource supply) would be positive and saturating in shape, while relationships between BEF proxies in background habitats (low-resource supply) would be positive and accelerating, and relationships between BEF proxies in the transitional 'chemotone' between these habitats (moderate-resource supply [44]) would be linear in shape [20] (figure 1). We thus theorized that ecosystem functioning in deep-sea chemosynthetic habitats may be more resilient to anthropogenic disturbance than ecosystem functioning in photosynthetically dependent deep-sea habitats.

## 2. Material and Methods

### (a) Faunal collection

Sampling of sediment macrofauna occurred on the Pacific margin of Costa Rica aboard the RV *Atlantis* using the submersible *Alvin* over May–June 2017 (cruise AT37-13) and October–November 2018 (cruise AT42-03), covering a depth range of 377–1908 m. Samples were collected using push cores from five methane seeps (Jaco Scar, Mound 12, Mound 11, Parrita Seep, and Quepos Landslide) and surrounding non-seep sediments (electronic supplementary material, figure S2).

In total, 76 cores (6.4 cm internal diameter, 10 cm depth) representing 38 distinct sampling points (two cores per sample) were collected and analysed. Annelid, peracarid crustacean and mollusc macrofauna (greater than 300 µm; 95.7% of macrofaunal individuals) were identified to species/morpho-species level [44].

Samples were categorized as representing 'active' chemosynthetic seep (13 samples; tissue $\delta^{13}$C = −33.38; s.e. = ±1.19 ‰), 'transition' (13 samples; tissue $\delta^{13}$C = −28.66; s.e. = ±1.09 ‰) or 'background' photosynthetically dependent (12 samples; tissue $\delta^{13}$C = −21.73; s.e. = ±0.98 ‰) environments based on visual indicators of seep activity and geochemistry data (hydrogen sulfide concentration, dissolved inorganic carbon (DIC) concentration and tissue $\delta^{13}$C of DIC) collected from associated push cores—see electronic supplementary material and Ashford *et al.* [44].

### (b) Quantifying biodiversity

Taxonomic and functional trait biodiversity were quantified using richness (number of species/functional trait categories), diversity (Shannon Index) and evenness (Pielou's Index) metrics in R v. 3.4.2 using the package 'vegan' [45,46] (table 1; electronic supplementary material, table S2). Functional trait biodiversity was quantified based on the scoring of taxa at the family level

**Table 1.** Details of proxies for ecosystem functioning, biodiversity metrics, oceanographic variables, terrain variables and chemosynthetic activity variables investigated in this study.

| variable grouping | variable(s) analysed |
| --- | --- |
| ecosystem functioning proxies | BPc metric, calcification metric, faunal density, faunal biomass |
| biodiversity | species richness (n), species diversity (Shannon Index), species evenness (Pielou's Index), functional trait richness (n), functional trait diversity (Shannon Index), functional trait evenness (Pielou's Index) |
| oceanographic variables | seafloor depth, seafloor temperature, seafloor salinity, seafloor oxygen concentration, export of surface productivity to seafloor |
| terrain | seafloor ruggedness, seafloor slope |
| terrain position index | seafloor terrain position index |
| chemosynthetic activity | activity category |

for 32 functional traits in seven groupings (electronic supplementary material, Dataset S1). Maximum association with a trait was scored a number '5', while no association with a trait was scored a number '0' [44]. Per sample, taxon-specific scores for functional traits (0–5) were multiplied by the abundance of that taxon and totalled across all taxa present, producing a sample (rows) by functional trait abundance (columns) matrix. Functional trait richness, diversity and evenness were calculated from this functional trait matrix using the same methodology as if the matrix was comprised sample species abundance.

### (c) Quantifying ecosystem functioning proxies

Ecosystem functioning was estimated based on four complementary proxies (table 1; electronic supplementary material, table S3). Standing stock (macrofaunal wet biomass per sample) was measured to 0.00001 g using an electronic balance (A&D GR-202) and converted to a per square metre value. Faunal density per sample was expressed as a number of individuals per square metre.

Community BPc was quantified following Queirós *et al.* [47] using the following equation:

$$BPc = \sum_{i=1}^{n} \sqrt{Bi/Ai} \times Ai \times Mi \times Ri.$$

where 'Mi' and 'Ri' refer to mobility and sediment reworking parameters specific to each family (electronic supplementary material, dataset S2), while 'Bi' and 'Ai' refer to the biomass and abundance of each family in a sample. Because we did not record biomass at the family level, we used average individual biomass per sample instead of species-specific values. When determining family mobility and sediment reworking potential, we used the mean value for all species within a family scored by Queirós *et al.* [47]. Families not scored by Queirós *et al.* [47] were scored based on literature and expert opinion (authors L.A.L., O.S.A. and G.W.R.).

Biogenic calcium carbonate content per sample was determined semi-quantitatively by scoring families for their relative degree of internal/external calcification (electronic supplementary material, dataset S2). Scorings were derived from literature

or expert opinion. Per sample, scorings were multiplied by the abundance of each family, and totalled across all families present. Please see electronic supplementary materials for additional methodological detail.

## (d) Quantifying environmental conditions

Point measurements of seafloor temperature, salinity, water depth and oxygen concentration were made by AUV *Sentry* during overnight transects of the seep sites and surrounding sediments. Values for environmental variables were interpolated to the resolution of *Sentry* bathymetry data (1 m$^2$) in QGIS 3.6.0 separately for each cruise sampling period. Values for each environmental variable for the year of sample collection were extracted at sampling locations in QGIS 3.6.0 (table 1; electronic supplementary material, table S4). Where *Sentry* data were unavailable, data from the CTD of HOV *Alvin* (SeaBird SBE49; temperature, salinity and depth) were substituted.

To quantify surface productivity overlying each sampling point, we obtained net primary productivity (NPP) data from the portal http://www.science.oregonstate.edu/ocean.productivity/custom.php. Global (1/12° resolution) monthly mean ocean NPP data were summed across the year immediately prior to each research cruise. Values at sampling locations were extracted in QGIS 3.6.0. To estimate the export of primary productivity to the depth of each sampling location, we applied the equations of Lutz *et al.* [48] (table 1; electronic supplementary material, table S4).

Seafloor slope, Terrain Ruggedness Index and Terrain Position Index were calculated in QGIS 3.6.0 based on *Sentry* bathymetry data using GDAL terrain analysis tools. Values at sampling locations were extracted in QGIS 3.6.0 (table 1; electronic supplementary material, table S4). Please see electronic supplementary materials for additional methodological detail.

## (e) Statistical analyses

The form of the relationship between biodiversity variables and ecosystem functioning proxies at active chemosynthetically dependent, transitional 'chemotone' and background photosynthetically dependent habitats was investigated via a generalized additive model (GAM) framework in R v. 3.4.2 using the package 'mgcv 1.8-33' [45,49]. Please note that GAMs reveal correlation only, and thus no conclusions can be made regarding causal relationships between the variables investigated. Here, ecosystem functioning proxies were modelled as the dependent variable, while biodiversity variables, and variables documenting facets of the environment, were modelled as independent variables. Prior to analysis, all continuous variables were standardized, missing environmental data (oceanographic variables and terrain variables (table 1); four values of 304 observations) were replaced with '0' after standardization, and dimensionality in the dataset was reduced by undertaking principal component analysis (PCA) in R v. 3.4.2. For ecosystem functioning proxies (dependent variable in models), the first two principal components captured 97.4% of the variance, while for taxonomic biodiversity and functional trait biodiversity (independent variables), the first two principal components captured 98.4 and 100.0% of the variance, respectively (tables 1 and 2). For oceanographic variables (independent variable), the first two principal components captured 99.5% of the variance, and for terrain variables (independent variable), the first principal component captured 99.5% of the variance (tables 1 and 2). Please see electronic supplementary material, table S5 for complete details of variable weighting on each PC axis.

Smoothers were specified for all continuous independent variables (electronic supplementary material, table S6) and optimized automatically using the generalized cross-validation criterion [49]. A different smooth was specified between BEF

**Table 2.** Variance explained, per variable grouping, by first and second principal components following a Principal Components Analysis.

| variable grouping | PC1 variance explained (%) | PC2 variance explained (%) | total (%) |
|---|---|---|---|
| ecosystem functioning proxies | 62.56 | 34.90 | 97.46 |
| taxonomic biodiversity | 78.82 | 19.58 | 98.40 |
| functional trait biodiversity | 62.09 | 37.88 | 99.97 |
| oceanographic variables | 71.64 | 27.84 | 99.50 |
| terrain | 99.50 | 0.50 | 100.00 |

proxies for each chemosynthetic activity category. Initial models took the following structure:

$$g(E(\text{Ecosystem functioning proxies } PC))$$
$$= \text{factor}(\text{Activity Category})$$
$$+ S_1(\text{Biodiversity } PC1, by = \text{Activity Category})$$
$$+ S_2(\text{Biodiversity } PC2, by = \text{Activity Category})$$
$$+ S_3(\text{Oceanographic } PC1) + S_4(\text{Oceanographic } PC2)$$
$$+ S_5(\text{Terrain } PC1) + S_6(\text{Terrain Position Index}), \varepsilon \sim N(0, \sigma 2).$$

Model performance diagnostics and the Akaike information criterion (AIC) were used to select appropriate error distributions and link functions [50]. GAMs were refined from initial full models via backward stepwise selection based on independent variable *p*-values and model AIC. The number of knots specified for each smoothed term was optimized based on model AIC and assumptions, with care taken to avoid overfitting (electronic supplementary material, table S6). Please see electronic supplementary material for additional details.

## 3. Results

## (a) Relationships between biodiversity and ecosystem functioning proxies by seep habitat category

Variance in ecosystem functioning proxies PC1 and PC2 was well explained by our models (adjusted model $R^2$ ranged between 0.28 and 0.68). As hypothesized, the form of relationship between BEF proxies differed among chemosynthetically dependent habitat, transitional 'chemotone' habitat, and photosynthetically dependent habitat. In chemosynthetic methane seep habitat, variance in both ecosystem functioning proxies PC1 and PC2 was high and significantly related to both taxonomic and functional trait biodiversity PC1 axes (ecosystem functioning proxies PC1: taxonomic biodiversity PC1, $p = 0.0015$; ecosystem functioning proxies PC1: functional trait biodiversity PC1, $p = 0.0001$; ecosystem functioning proxies PC2: taxonomic biodiversity PC1, $p = 0.0104$; ecosystem functioning proxies PC2: functional trait biodiversity PC1, $p = 0.0008$) (table 3). However, contrary to our hypothesis, these relationships were unimodal in form; a peak in ecosystem functioning

**Table 3.** Significant relationships between ecosystem functioning proxies PC axes (dependent variables) and continuous independent variables in GAM constructed.

| dependent variable | independent variable | F-value | p-value |
|---|---|---|---|
| ecosystem functioning proxies PC1 | taxonomic biodiversity PC1 (active) | 5.723 | 0.0015 |
| ecosystem functioning proxies PC1 | taxonomic biodiversity PC2 (active) | 6.914 | 0.0009 |
| ecosystem functioning proxies PC1 | functional trait biodiversity PC1 (active) | 6.859 | 0.0001 |
| ecosystem functioning proxies PC1 | functional trait biodiversity PC2 (active) | 3.264 | 0.0006 |
| ecosystem functioning proxies PC2 | taxonomic biodiversity PC1 (active) | 5.005 | 0.0104 |
| ecosystem functioning proxies PC2 | oceanographic PC2 | 2.907 | 0.0154 |
| ecosystem functioning proxies PC2 | functional trait biodiversity PC1 (active) | 9.047 | 0.0008 |
| ecosystem functioning proxies PC2 | terrain PC1 | 2.182 | 0.0278 |

proxies typically occurred around mean values of biodiversity, (figure 2). Relationships between ecosystem functioning proxies PC1 and PC2 and taxonomic and functional trait biodiversity PC2 were weaker (ecosystem functioning proxies PC1: taxonomic biodiversity PC2, $p = 0.0009$; ecosystem functioning proxies PC1: functional trait biodiversity PC2, $p = 0.0006$; ecosystem functioning proxies PC2: taxonomic biodiversity PC2, $p = 0.4029$; ecosystem functioning proxies PC2: functional trait biodiversity PC2, $p = 0.5955$) (table 3). Where significant relationships were found, these were either unimodal in form (peak in ecosystem functioning proxies at around mean values of taxonomic biodiversity), or positive linear (ecosystem functioning proxies increase linearly with functional trait biodiversity) (figure 2).

In the transitional 'chemotone' between chemosynthetic and photosynthetic deep-sea habitats, variance in ecosystem functioning proxies PC1 and PC2 was smaller than in active seep habitat, and contrary to our hypothesis, no significant relationships with taxonomic or functional trait biodiversity were uncovered (table 3 and figure 2).

In lower resource photosynthetically dependent background habitat, variance in ecosystem functioning proxies PC1 and PC2 was small, and again, contrary to hypotheses, no significant relationships with taxonomic or functional trait biodiversity were uncovered (table 3 and figure 2).

## (b) Influence of environment on ecosystem functioning proxies

Ecosystem functioning proxies varied significantly with oceanographic variables PC2, and terrain PC1, but not with oceanographic variables PC1 or terrain position index (table 3). Ecosystem functioning proxies PC2 varied with oceanographic variables PC2 ($p = 0.0154$) (table 3). This relationship was unimodal; peak values of ecosystem functioning proxies occurred at intermediate values of oceanographic variables PC2, corresponding closely to the oceanographic conditions observed at 'Mound 11' and 'Mound 12'. Ecosystem functioning proxies PC2 varied significantly with terrain PC1 ($p = 0.0278$); maximal values of ecosystem functioning proxies PC2 occurred in areas with high ruggedness and slope.

## 4. Discussion

With biodiversity in decline worldwide [8], knowledge of the factors that influence the form of BEF relationships is crucial

to our understanding of the relationship between biodiversity loss and ecosystem functioning. Here, undertaking a mensurative study, we used methane seepage gradients as a natural laboratory in which to explore how the strength and form of relationship between BEF proxies is influenced by changing trophic resource characteristics in a little-studied realm that is under increasing threat of disturbance from human resource extraction.

Elements of our results both conform with and contradict our hypotheses, but point to a potential need for alternative management strategies in methane seeps relative to photosynthetically dependent deep-sea ecosystems. The finding of strong, unimodal relationships between BEF proxies in high-resource supply methane seep habitats (figure 2) contradicts our initial hypothesis. That the form of these relationships were quite consistent in this habitat across different BEF proxies PC axes suggests that the shared characteristics of these axes may be important in determining the overall form of the relationships investigated. For biodiversity variables, species evenness and functional trait richness mapped similarly to axes PC1 and PC2, while for ecosystem functioning proxies, the degree of calcification and faunal density mapped similarly to axes PC1 and PC2. This suggests an important relationship between species evenness and functional trait richness, and degree of calcification and faunal density in methane seep habitats. These patterns may reflect associations between microbial and animal components of the habitat and the physiological pressures imposed (such as high hydrogen sulfide and low oxygen concentrations in sediment pore waters). Depressed proxies for ecosystem functioning at low levels of biodiversity may correspond to the most active areas of seeps, where overwhelming physiological pressures dominate. A peak in proxies for ecosystem functioning at moderate levels of biodiversity may be linked to an optimal even assemblage of specialist species with particular functional traits that are positively associated with ecosystem functioning in environments necessitating seep-specific adaptations, such as the animal/microbe symbioses known in some species of seep invertebrate [51,52]. Other research at Costa Rican methane seeps suggests that these specialist macrofaunal species may include cumaceans and bivalve and gastropod molluscs [44], all of which have a relatively high degree of calcification. A decline in ecosystem functioning proxies alongside high biodiversity may reflect dilution of optimal assemblages by less-specialized taxa in more benign seep habitats, and competition for resources among seep-specialist and non-seep obligate taxa.

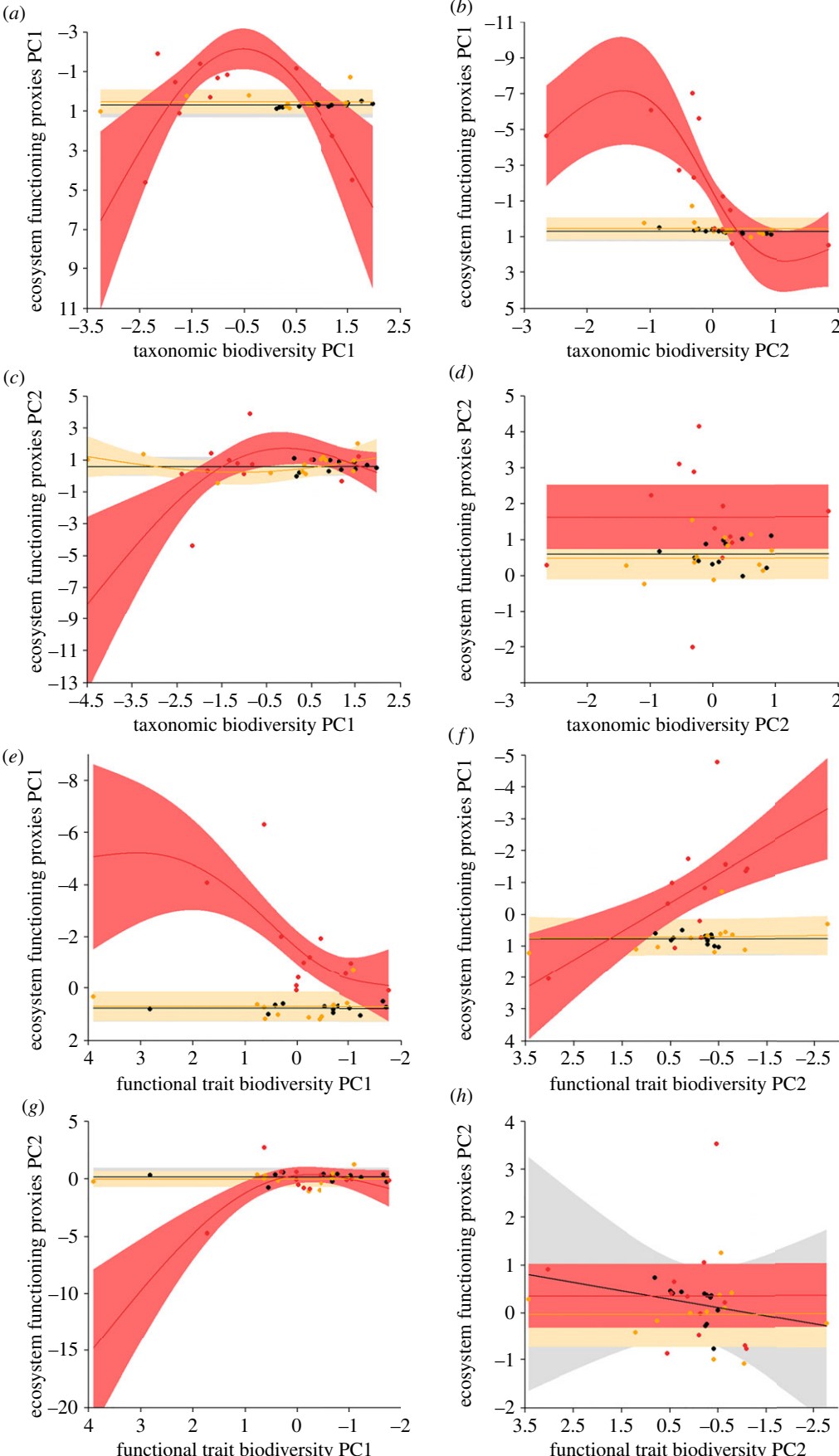

**Figure 2.** Relationships between BEF proxies obtained for different deep-sea habitats. (*a*) Taxonomic biodiversity PC1 versus ecosystem functioning proxies PC1; (*b*) taxonomic biodiversity PC2 versus ecosystem functioning proxies PC1; (*c*) taxonomic biodiversity PC1 versus ecosystem functioning proxies PC2; (*d*) taxonomic biodiversity PC2 versus ecosystem functioning proxies PC2; (*e*) functional trait biodiversity PC1 versus ecosystem functioning proxies PC1; (*f*) functional trait biodiversity PC2 versus ecosystem functioning proxies PC1; (*g*) functional trait biodiversity PC1 versus ecosystem functioning proxies PC2; (*h*) functional trait biodiversity PC2 versus ecosystem functioning proxies PC2. Solid lines are lines of best fit as determined by GAM. Shaded bands are 95% confidence intervals; black line, grey band: background habitat; orange line, beige band: transitional 'chemotone' habitat; red line, light red band: active seep habitat. Note: ecosystem functioning proxies PC1 axis and functional trait biodiversity PC1 and PC2 axes are reversed on figures to ease interpretation (see electronic supplementary material, table S5). (Online version in colour.)

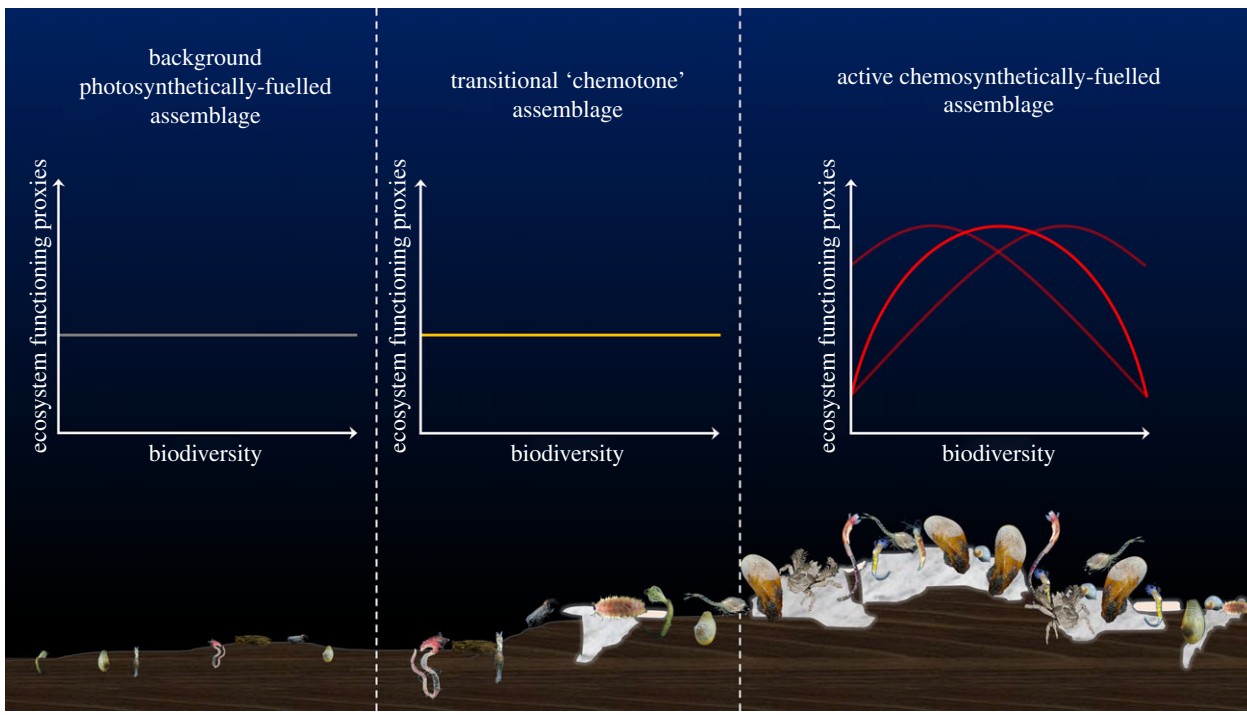

**Figure 3.** Conceptual diagram illustrating how BEF relationships may change along a gradient of resource availability. Photosynthetically dependent background habitat: grey line, no relationship; transitional 'chemotone' habitat: orange line, no relationship; chemosynthetically dependent methane seep habitat: red line, unimodal form, the peak of which may be pushed to either higher or lower biodiversity values. (Online version in colour.)

That we did not find evidence for increasingly positive relationships between biodiversity and proxies for ecosystem functioning in background deep-sea habitats (figure 2) is in contrast with our predictions and the findings of several studies [18,22,23,53,54]. Many of these studies emphasize the importance of facilitation in structuring deep-sea ecosystems [4,18,21]. For example, behaviours exhibited by some deep-sea polychaete species, such as the rapid subduction of fresh detrital material, the construction of tubes and burrows, and the bioturbation of sediments through movement, may promote greater faunal and microbial diversity and ecosystem functioning by increasing structural complexity, habitat space, resource heterogeneity and availability, and altering the timing and location of organic matter burial [55–57]. Our failure to uncover relationships between biodiversity and proxies for ecosystem functioning in photosynthetically dependent background habitats may reflect the subtle nature of these relationships and represent a type II error associated with our relatively small sample size.

Also contrary to our initial hypotheses, we did not find evidence for a linear relationship between BEF proxies in the 'chemotone' between chemosynthesis- and photosynthesis-dependent habitats (figure 2). Made possible by the breadth of the chemotone at seeps, relationships between BEF proxies in this zone may reflect a dynamic interaction of opposing relationships between the unimodal form characteristic of methane seeps (associated with physiological constraints, seep specialization and competition), and the accelerating form that has been shown to be characteristic of background photosynthetically dependent deep-sea habitat. This possible dynamic interaction of opposing methane seep and background ecological forces, alongside the relatively low statistical power of our study, may have contributed to our failure to identify relationships between BEF proxies in this environment.

Our results provide evidence that relationships between BEF proxies in chemosynthetic methane seep habitats are distinct from those of more resource-limited (with respect to availability and diversity) background deep-sea environments dependent on photosynthetic surface production. It is possible that unimodal relationships between BEF proxies are characteristic of marine chemosynthetic environments more generally, where faunal densities, standing stock and the proportion of calcifying taxa may be high, but species evenness and diversity may be relatively low [41,58,59]. Future studies should investigate the relationship between BEF in other marine chemosynthetic environments, such as hydrothermal vents.

In agreement with past studies [27,30,32,34,60], our results suggest that the influence of biodiversity on ecosystem functioning may be dependent upon the characteristics of available resources, among other factors (electronic supplementary material, table S1). Our results demonstrate that relationships between BEF may not always be positive, and indeed, suggest that in circumstances where high-resource supply is associated with a physiologically stressful or toxic environment, a peak in ecosystem functioning may occur when biodiversity is only moderate [38]. This suggests that biodiversity, as reflected in numbers and evenness of species or traits, should not necessarily be used as a direct metric of ecosystem 'value' or 'health' in these environments. Mangrove, salt marsh and seagrass sediments represent similar environments to methane seeps, where high-resource availability is associated with physiologically harsh conditions [61,62], and, in some cases, with chemosynthetic production [63]. It may be insightful to investigate the form of the relationship between BEF proxies in these habitats.

Considering our results holistically, one can envision a situation whereby moving from high-resource supply but physiologically stressful methane seep environments, towards low-resource supply, but less stressful background deep-sea habitats, relationships between BEF proxies change from a unimodal form, to a positive accelerating form, with a transition between these states in the 'chemotone' (figure 3). This variation in form of the relationship with changing habitat suggests that

biodiversity loss in contrasting deep-sea environments may be correlated with differing changes in ecosystem functioning. Lower and higher resource deep-sea environments may require different management strategies to maintain ecosystem functioning under the threat of human disturbance. Prior studies suggest that in resource-poor deep-sea environments, even a small biodiversity loss may be associated with a relatively large decline in ecosystem functioning [18,22,23,53,54] (electronic supplementary material, figure S1). Strong measures to protect biodiversity in these low-resource environments may be key to maintaining ecosystem functioning. By contrast, in high-resource chemosynthetically dependent environments, a more limited set of specialized species may be important in supporting high levels of ecosystem functioning. Effective management of these environments may require a strong focus on the protection of critical species, as opposed to maximizing biodiversity outright [64].

Human activities have already disturbed large areas of the deep sea, particularly along continental margins [65–67], and ecosystem functioning in some areas of the deep ocean may already be depressed below natural levels. Deep seabed mining represents an emerging and serious threat to a variety of deep-sea ecosystems, including resource-poor, nodule covered abyssal plains and resource-rich chemosynthetically dependent hydrothermal vents [66]. Methane seeps are vulnerable to disturbance from oil spills, oil and gas extraction, and potentially gas hydrate exploitation [68,69]. Planned extraction activities in resource-poor environments, such as the Clarion–Clipperton fracture zone (CCFZ) may result in a significant decline in ecosystem functioning, even if effective management procedures limit the absolute magnitude of biodiversity loss [70,71] (electronic supplementary material, figure S1). The lower alpha diversity of resource-rich methane seep environments [41,58,59] coupled with the potential association of a specialized selection of species with ecosystem functioning imply that the loss of a small number of these critical species could be correlated with a major decline in ecosystem functioning. Although 'biodiversity hotspots' typically receive the greatest attention when it comes to conservation strategies [72], conservation efforts and management planning in methane seeps should concentrate on 'functioning hotspots' [73].

The greatly contrasting relationships between BEF proxies obtained here for habitats in close spatial proximity to one another emphasizes that it is difficult to generalize these relationships. While the deep ocean is often characterized as a vast, homogeneous habitat, our results depict heterogeneous environments in close proximity to one another that are fuelled by different resources and governed by different ecological relationships [40,74].

**Ethics.** E.E.C. and J.C. Fieldwork was carried out under permits granted by the Costa Rica Ministerio de Ambiente y Energía (Sistema Nacional de Áreas de Conservación/Comisión Nacional para la Gestión de la Biodiversidad): cruise AT37-13: SINAC-CUS-PI-R-035-2017; cruise AT42-03: SINAC-SE-064-2018. This research was performed in accordance with all applicable international, national and/or institutional laws, guidelines and ethical standards.

**Data accessibility.** All data used by this study are freely available in the electronic supplementary material [75], and on the Biological & Chemical Oceanography Data Management Office (BCO-DMO) website: https://www.bco-dmo.org/project/648472.

**Authors' Contributions.** O.S.A.: conceptualization, data curation, formal analysis, funding acquisition, investigation, methodology, project administration, resources, software, supervision, validation, visualization, writing-original draft, writing-review and editing; S.G.: investigation, methodology, writing-review and editing; D.C.: investigation, methodology, writing-review and editing; K.R.: investigation, methodology, writing-review and editing; K.R.: investigation, methodology, writing-review and editing; G.M.: methodology, writing-review and editing; E.E.C.: data curation, funding acquisition, project administration, resources, writing-review and editing; J.C.: funding acquisition, writing-review and editing; G.R.: funding acquisition, methodology, resources, writing-review and editing; A.K.S.: conceptualization, writing-review and editing; L.L.: conceptualization, funding acquisition, investigation, methodology, project administration, resources, supervision, writing-review and editing.

All authors gave final approval for publication and agreed to be held accountable for the work performed therein.

**Competing interests.** The authors declare no competing interests.

**Funding.** This work was funded by National Science Foundation Ocean Sciences grant nos. 1634172 and 1635219. Author O.S.A. additionally received support from Scripps Institution of Oceanography as a Post-doctoral Scholar.

**Acknowledgements.** We thank the Costa Rica Ministerio de Ambiente y Energía (Sistema Nacional de Áreas de Conservación/Comisión Nacional para la Gestión de la Biodiversidad) for granting collection permits (AT37-13: SINAC-CUS-PI-R-035-2017, AT42-03: SINAC-SE-064-2018), and we thank the Captains, Crew, and *Alvin* team of RV *Atlantis* legs AT37-13 and AT42-03 for facilitating sample collection. We are grateful to Jennifer Le, Odalisca Breedy, Lillian McCormick, Natalya Gallo and Olivia Pereira for their help with sectioning and preserving samples. We are grateful to Professor Moriaki Yasuhara and three anonymous reviewers, whose insightful and constructive comments helped to strengthen this article.

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
