## [Peer Review File · Proceedings of the Royal Society B: Biological Sciences]

Review History

RSPB-2020-0828.R0 (Original submission)

Review form: Reviewer 1 (Moriaki Yasuhara)

Recommendation

Accept with minor revision (please list in comments)

Scientific importance: Is the manuscript an original and important contribution to its field?

Excellent

General interest: Is the paper of sufficient general interest?

Excellent

Quality of the paper: Is the overall quality of the paper suitable?

Excellent

Is the length of the paper justified?

Yes

Should the paper be seen by a specialist statistical reviewer?

No

Do you have any concerns about statistical analyses in this paper? If so, please specify them explicitly in your report.

No

It is a condition of publication that authors make their supporting data, code and materials available - either as supplementary material or hosted in an external repository. Please rate, if applicable, the supporting data on the following criteria.

Is it accessible?

Yes

Is it clear?

Yes

Is it adequate?

Yes

Do you have any ethical concerns with this paper?

No

Comments to the Author

BEF has been rarely studied in deep sea and never studied in chemosynthetic system despite of its importance as a promising model system for BEF research.

It's partly because quantification and robust estimation of ecosystem function is not easy there. This paper indeed did what we needed in BEF study and deep-sea biology in very high quality with multiple robust measures of biodiversity and ecosystem functioning, using samples from Costa Rican cold seeps. The result is very clear and makes sense. It's neat study and I enthusiastically recommend publication after minor revision.

My comments are mostly minor and detailed below. The manuscript is well written and clear, but can be a bit more improved by simplifying the terminology and reducing redundancy (I am sure it can be done easily by the authors).

Line 62-63: The author considers normal deep-sea is still photosynthesis-based, because they rely on POC flux? I think it's good to explain this sort abit for non-deep-sea biologist audience here or somewhere.

Later parts of introduction is already Methods.

Introduction can be shorten in this regard.

'Background' photosynthetically-fuelled samples still have quite low $\delta^{13}\text{C}$ (-21.73) compared with normal sediment. Some comment needed on it?

Font size small in fig 2.

The authors every time say chemosynthetically-fuelled habitat, photosynthetically-fuelled habitat, and the transition zone. You may define in the first appearance and just say seep, non-seep and transitional (or active, background and transition as in Table 2) thereafter wherever details don't matter?

Lines 264-266. Better to combine into one sentence.

Line 285. species diversity means Shannon? May be good to up it in brackets.

The paragraph of Line 327 or somewhere in the discussion: It's good to mention environmental (=oceanographic characteristics) control of biodiversity in non seep system is consistent with our understanding, by citing, eg Sweetman et al 2017 Elementa, Yasuhara and Danovaro 2016 Biological Reviews.

Line 429 on functional hotspots: Logic is not super clear, because the author's actual result indicate high-diversity non-seep system is more vulnerable from species loss to undermine ecosystem function.

But here in this paragraph the author argue almost opposite "loss of a small number of species could result in the collapse of ecosystem functioning" in seep system.

I think it's good to re-frame the discussion and flow of this paragraph.

Chapman et al 2019. sFDvent: A global trait database for deep-sea hydrothermal-vent fauna. Global Ecology and Biogeography: 28, 1538–1551.
This paper is worth mentioning somewhere.

Good to have locality map somewhere.

Moriaki Yasuhara (Signed review)

Review form: Reviewer 2

Recommendation

Accept with minor revision (please list in comments)

Scientific importance: Is the manuscript an original and important contribution to its field?

Good

General interest: Is the paper of sufficient general interest?

Good

Quality of the paper: Is the overall quality of the paper suitable?

Excellent

Is the length of the paper justified?

Yes

Should the paper be seen by a specialist statistical reviewer?

Yes

Do you have any concerns about statistical analyses in this paper? If so, please specify them explicitly in your report.

Yes

It is a condition of publication that authors make their supporting data, code and materials available - either as supplementary material or hosted in an external repository. Please rate, if applicable, the supporting data on the following criteria.

Is it accessible?

Yes

Is it clear?

Yes

Is it adequate?

Yes

Do you have any ethical concerns with this paper?

No

Comments to the Author

I would like to thank the authors for producing such an interesting manuscript with such a thorough explanation of key concepts and methodological processes. I really appreciated the opportunity to read and review this manuscript. Here, the authors use deep-sea methane seeps as a natural laboratory, to investigate biodiversity-ecosystem functioning (BEF) relationships along an energy gradient (from high-resource seep to low-resource background communities). Furthermore, they use Structural Equation Modelling to investigate possible environmental drivers influencing the form of BEF relationships in different communities. The aim of the manuscript is, therefore, to test the hypothesis that the form of BEF relationships will change from background, photosynthetically fuelled deep-sea habitats to chemosynthetically fuelled methane seeps, before identifying drivers behind any differences identified. This differs from previous research in two main ways: i) habitat, wherein methane seeps and surrounding seafloor habitats provide a measurable energy gradient to test such a hypothesis and have not been examined in such a way before; and ii) investigating drivers of BEF relationship form beyond topographic characteristics of the seafloor. This manuscript therefore makes an important contribution to the BEF literature, whilst placing methane seeps into a wider ecological literature, making a positive step forward in introducing these sometimes deemed 'non-traditional' systems to ecologists familiar with more 'traditional' ones as examples of testbeds for ecological theory. I comment on each section of the manuscript below, before making one major comment and more minor ones. I hope that these prove useful.

Title

The title of this manuscript is clear and succinct, explaining the key finding well. I would suggest that the authors consider if there is a way to make it appealing to wider audiences, however, if there are general BEF and energy relationships that might also be present in other ecosystems. I propose this as I found this a very interesting paper and feel it could appeal to a wide range of biologists and ecologists, but might not get 'picked up' as often because of the specification of chemosynthetic systems in the title. It appears, based on the discussion, that the management finding might also be applicable in other physiologically demanding systems, so perhaps 'physiologically challenging systems' could be a good option.

Abstract

The abstract of this article is clear and concise, providing good insights into the approach and key findings of this work. I wonder whether, instead of the take-home message that 'photosynthetically-fuelled and chemosynthetically-fuelled environments should be managed differently', it might also be good to highlight differences in energy/resources which might also apply to other ecosystems.

Introduction

The introduction is very clear. I appreciated the explanations of the different BEF relationship forms under different resource availabilities and this set me up well as a reader to understand your rationale for your hypotheses. I make some suggestions regarding Figure 1 under minor comments, to try to ensure it further complements this well-written text.

Materials and Methods

Section 3.1 of the Materials and Methods section is very detailed and well-explained. I really appreciate this, as I have not sampled seeps, yet could follow the rationale and processes. Section 3.5 is also detailed and clear, with helpful Supporting Information. Contrastingly, I feel section 3.2 could benefit from additional detail beyond that provided in Supporting Information (see Major Comments). For instance, 32 is a rather large number of traits, with impacts on the outcomes of diversity measures if pooled into a multidimensional metric (hence most trait-based studies focusing on ~5 traits). I was also left wondering what the sample by trait matrix was used as an input for, so this needs to be explained. Whilst I have trait-based expertise, I struggled to follow the rationale for this part of the analysis and fear those who aren't familiar with trait-based methods would need even more detail than the extra information I am looking for.

Results

The results are clear and concise and well-supported by further details and re-iteration in the Discussion.

Discussion

I would recommend starting the Discussion section with your main finding(s), rather than re-setting the scene, though I feel the introductory paragraph is strong and the text could be used later in this paragraph and/or moved to the Introduction. I found the Discussion interesting and particularly liked some of the examples used to explain some of the patterns found (e.g. the polychaetes and dominant species). I also found the interpretation regarding a need for investigation in other high resource, physiologically harsh environments compelling and hope that this will fuel future research. Finally, I appreciated the discussion centring on management strategies, as it might not be immediately obvious to readers that there is a clear conservation-oriented message in the findings without this written support.

Supporting Information

The Supporting Information is detailed and complements the work presented in the main manuscript. I appreciated the different forms of the SEMs, as well as the literature review summary, for reference. I make comments on Table S2 under 'Minor Comments' and these relate to the 'Major Comment' below. I really appreciated the provision of trait scoring and references.

Major Comments:

1. My only major comment relates to the use of functional traits in this manuscript, as I find the methods difficult to follow (e.g. I am unclear as to which metrics have been used and why, even after reading the Supporting Information). I appreciated the provision of trait data in the Supporting Information, but struggled to understand how these were processed and used (e.g. it is stated that a suitable matrix is produced but I am unclear as to how this was then used). I was also concerned by the inclusion of 32 traits, which seems to me to be too many (as, if traditional trait-based diversity indices have been used, having so many traits will, essentially, produce values as if every species is unique, thereby almost replicating taxonomic measures). I therefore recommend that more detail is added to the methods (see Minor Comments below) and sensitivity testing is added to Supporting Information, or the trait-based approach is removed from this paper, as I am not clear on how it adds substantially to the testing of the hypothesis set out in the Introduction. I am reluctant to make the latter of these suggestions, as I appreciate the use of a trait-based approach to complement 'traditional' measures of biodiversity, but I am concerned that the approach has not been tested sufficiently for me to be confident in the outcomes of this part of the analysis. It also seems that many of the discussion points centre on specifically measured characteristics and do not depend on the multivariate indices (e.g. lines 330-338 focus in on bioturbation and lines 344-348 highlight an interesting role of dominant

species).

Minor Comments:

Line 27: Propose adding '-' after relationships, for consistency with the punctuation in the rest of the sentence.

Line 29: I'm not sure how true this statement is, particularly given the wealth of studies investigating biodiversity and ecosystem functioning relationships in plants. It might be 'safest' to specify 'marine' photosynthesis-based or chemosynthesis-based ecosystems, rather than stating that BEF relationships haven't been explored in detail for photosynthesis-based ecosystems. This is particularly important to consider given the statements in lines 57-60, which suggest that photosynthesis-based ecosystems have been well-studied.

Line 30: I propose adding reference to methane seepage gradients as 'natural laboratories', in line with the main text, as I think this is a great way to highlight deep-sea ecosystems as systems to be considered in a wealth of ecological work.

Line 32: I propose removing 'trait', as 'functional diversity' is more commonly used terminology, or 'trait-based diversity'.

(Lines 38-39: I highlight this sentence as I think it provides a particularly nice summary on the importance of a key finding of this study. I therefore think it would make a good sentence for further communication associated with this work and wanted to 'flag' it as a sentence to 'keep' and re-use (e.g. for research end-user / engagement activities).)

Line 45: I propose changing 'monetarily' to 'economically', as I notice as a reader that this word could easily be mistaken for 'momentarily', if read in a rush, which would be an issue.

Line 46: I propose changing 'extremely important' to 'vital', or another single-word descriptor.

Line 49: I propose changing 'services' to 'contributions' (see comment on line 54 below).

Line 54: A minor comment here that some readers/reviewers might expect the term 'ecosystem services' to be updated given recent IPBES reporting and surrounding discussions terming these 'nature's contributions to people' (e.g. see: <https://www.tandfonline.com/doi/full/10.1080/26395916.2019.1669713#:~:text=In%202017%20the%20term%20was,2017>). It might also be helpful to add a word or two to line 53 to ensure that the link between ecosystem functioning and ecosystem services is clear (e.g. 'in promoting the functioning of ecosystems which, ultimately provide important services to humanity', or an alternative acknowledging the discussions regarding services vs. contributions).

Line 77: I propose changing the word 'expressed', as this might make the reader think of genetic traits, whereas functional traits are characteristics of a species.

Line 83: I wonder if it might be best to just refer to methane seeps (not vents), as this is the focus of this paper and they are a great example on their own (so introducing vents but not including them in your analyses might add unnecessary confusion for readers unfamiliar with chemosynthesis-based ecosystems).

Lines 189-190: I feel that the impact of this choice should be tested for in supplementary methods (e.g. changing the average and testing for the influence of this being 'off' for some species). Alternatively, there needs to be a justification as to why it is OK to use average individual biomass per sample instead of species-specific values, as this sounds like a limitation, but it is unclear as to whether it is and, if so, why it is an acceptable one or what could be done to

minimise the limitations on the results.

Lines 192-193: 'the opinion of authors' could be summarised as 'expert opinion' (with initials in brackets, as these are helpful for anyone wanting to contact these authors), to ensure that those less familiar with the deep-sea research community understand the 'expert' status of these authors with respect to trait scoring (i.e. I know that these authors are definitely experts, with so much knowledge and experience, but a terrestrial ecologist working on reptiles might not, for example).

Line 211: I propose adding a few words justifying the choice of inverse-distance weighting over other interpolation methods.

Line 283: Trait evenness might be capturing the dominance of certain species in high-resource areas, if evenness increases with resource.

Line 344-345: I just wanted to add a comment that I found similar indications in trait-based studies of hydrothermal-vent ecosystems, so find this particularly interesting as it seems to follow what is also seen at vents.

Lines 355-356: I'm not sure the reference to hydrothermal vents in line 355 is well-placed, as diversity can vary and evenness can be very high at vents, due to the dominance of certain species (contradicting your statement in line 356).

Lines 360-363: Could this also be because the hypothesis, or previous work, is based on an intermediate disturbance hypothesis, whereas seeps have more of a steady gradient (e.g. not a large, abrupt transition like at a vent chimney)?

Lines 408-412: This is an important implication of this research, which I feel could be good to explain a little further in the abstract to capture readers from across disciplines and ecological realms (e.g. the distinction between whole-community conservation and critical species conservation).

Lines 423-426: I feel this is another critical point that should be made earlier in the discussion (e.g. this part could be moved up).

Table 1: I think the metrics 'trait richness, trait diversity, trait evenness' need to be specified in the methods, as well as any package(s) used, as it can influence the results.

Figure 1: This is a helpful figure, complementing the descriptions in the introduction. However, I would recommend adding labels to the lines to further complement the written text, so a reader can instantly see which habitat/hypothesis each line represents (as described in the caption) and some of the key ecological conditions one might expect each type of BEF relationship to occur under (as described in the paragraph starting on line 80).

Figure 2: This is very minor but please consider increasing the font size in this figure, as it is very hard to read.

Figure 3: One data point seems to be driving the shape of plots A and C (e.g. both could be straight without the left-hand points). I therefore feel reference should be made to the supplementary information containing alternative runs, where different options for the setup of the SEM were investigated. Please also consider increasing the font sizes, as this figure is very difficult to read.

Figure S1: I think this is an important supporting figure, but feel the caption would benefit from more detail on what saturating, linear, and accelerating mean, so readers don't need to flick

between this and the main manuscript, if possible.

Table S1: This is very helpful. It might be worth checking whether work by Martin Solan, Jasmin Godbold, and colleagues is relevant to cite here (as I understand they have conducted studies into BEF relationships but sometimes found no relationship, which should also be reported on here, if I have remembered this correctly).

Table S2: I'm afraid I'm not sure I understand your description of functional trait richness, as it differs from examples that I have seen, so needs further explanation and justification. This is also the case for 'functional trait diversity'. I have not seen this particular version of this measure in the trait-based literature, so it would be good to know why this approach has been used and how the results compare when using more typically used measures of functional diversity. I am also confused by the description of 'functional trait evenness', as there is a specific functional evenness metric and I'm not sure how this one compares.

Review form: Reviewer 3

Recommendation

Major revision is needed (please make suggestions in comments)

Scientific importance: Is the manuscript an original and important contribution to its field?

Good

General interest: Is the paper of sufficient general interest?

Excellent

Quality of the paper: Is the overall quality of the paper suitable?

Good

Is the length of the paper justified?

Yes

Should the paper be seen by a specialist statistical reviewer?

No

Do you have any concerns about statistical analyses in this paper? If so, please specify them explicitly in your report.

Yes

It is a condition of publication that authors make their supporting data, code and materials available - either as supplementary material or hosted in an external repository. Please rate, if applicable, the supporting data on the following criteria.

Is it accessible?

No

Is it clear?

N/A

Is it adequate?

N/A

Do you have any ethical concerns with this paper?

No

Comments to the Author

This is a very interesting paper that examines biodiversity – ecosystem functioning relationships along a energy input gradient in deep sea communities. My review is primarily focused on the statistical approaches. I can say though, as a terrestrial ecologist, I see direct applicability of the conceptual results to my study systems. I also appreciate the clarity of the writing.

I do have some concerns about the statistical approach. I see none of these as “dealbreaker” issues, but some revision is needed. In some cases, some additional explanation and justification of the approaches taken may be sufficient. In other cases, the authors may want to make revisions to the analyses based on these suggestions.

Most importantly, I was surprised to see that the authors chose to divide their analyses into 5 separate models. I am not sure that this was the right decision for a number of reasons. First, given the relatively low sample sizes, this seems to be cutting these data rather fine. Second, conceptually the authors are treating these sites as a gradient of energy input (Figure 4). Given that they are conceiving these as a gradient, why not analyze as such. Finally, there does not seem to be a compelling practical reason for the separation as the statistical approach can readily handle the nonlinearity.

Sample Size. I recognize that there are substantial logistical limitations to sample sizes in this research area. I am concerned, however, that many of the curvilinear relationships shown in figure 3 are driven by single datapoints. Panels A and C would be linear without the low biodiversity points. That said, I don't think this is a huge problem as the general conclusions from those panels would not be different substantially different minus the low points.

Latent variable selection: The biodiversity latent includes richness, evenness, and diversity measures. Given that evenness and richness measure very different things, are often structured by different ecological factors, and are often uncorrelated, a strong justification is needed to combine them into a single latent variable. For a plant example, see this paper showing how evenness and richness are structured by very different processes in a plant community. Wilsey, B. J., D. R. Chalcraft, et al. (2005). "Relationships among indices suggest that richness is an incomplete surrogate for grassland biodiversity." *Ecology* 86(5): 1178-1184

I would encourage the authors to replace the initial SEM figure in the main text with the fitted figures (S3/4), as those figures are showing the results. Putting dotted lines in to represent NS relationships removed from the models would be a good way of showing the initial SEM form. I would also encourage some overall fit statistics to be placed on S3/4 when it is moved to the main text.

Overall figure S3/4 could be made much more compact to place all 5 panels together. You may also want to consider ovals to surround latent variables. Many SEM applications (see papers by Grace for examples) by convention use ovals for latent variables and rectangles for directly observed variables.

As a terrestrial plant ecologist, I find figure 4 curious because I normally think of productivity as an ecosystem function. I wonder if characterizing these as high energy input and low energy input systems would be a better approach.

L247. Simply replacing missing values with the mean is a pretty simplistic way of dealing with missing data. Given that strong correlations are likely among environmental values, a more sophisticated approach to estimating these values could easily be used.

Review form: Reviewer 4

Recommendation

Reject – article is scientifically unsound

Scientific importance: Is the manuscript an original and important contribution to its field?

Good

General interest: Is the paper of sufficient general interest?

Good

Quality of the paper: Is the overall quality of the paper suitable?

Poor

Is the length of the paper justified?

Yes

Should the paper be seen by a specialist statistical reviewer?

Yes

Do you have any concerns about statistical analyses in this paper? If so, please specify them explicitly in your report.

Yes

It is a condition of publication that authors make their supporting data, code and materials available - either as supplementary material or hosted in an external repository. Please rate, if applicable, the supporting data on the following criteria.

Is it accessible?

Yes

Is it clear?

No

Is it adequate?

No

Do you have any ethical concerns with this paper?

No

Comments to the Author

I was asked to act as a specialist statistical referee for manuscript RSPB-2020-0828 titled “Biodiversity affects ecosystem functioning differently in chemosynthetic systems invoking new management focus” authored by Ashford et al. I have experience in multi-level modeling and structural equation modelling that makes me a suitable referee for this manuscript. In addition, because a significant focus of my research program is to explore causal relationships between biodiversity and ecosystem function (albeit in very different ecosystems), I also feel qualified to address the appropriateness of statistics and models to answer the paper’s core proposed research questions.

The authors provided access to three datasets, but the one pertinent to my review and was used to try and recreate the results was ‘Dataset_S3.xlsx’. I appreciate that the authors made their data available, which is key to producing reproducible research. However, there were many mismatches between the variable names in the manuscript and the dataset which hindered reproducibility. These were often simple things that could be fixed with some renaming and

reorganization. For example, indicators of the 'Biodiversity' latent variable were referred to as 'trait richness', 'trait diversity', and 'trait evenness', but in the data table they were named 'Functional_richness', 'Functional_diversity', and 'Functional_evenness' (at least the reader is left to assume that these are the same variables). So, regardless of the fate of this manuscript I have several data management and presentation suggestions that will aid in making results reproducible:

- (1) Rename the variables in the dataset and to match descriptions in the main text. Use short, meaningful column names rather than variable names such as: "Seafloor_POC_concentration_for_sampling_year_and_previous_year_mg/m⁻³/month", which is actually the name of one of the columns in Dataset_3.xlsx. A much better choice would have been 'POC'.
- (2) Include a metadata file that lists every variable in Dataset_S3.xlsx and provides information about each variable, the units, etc. So, for example, for the 'POC' variable one could then list "concentration for sampling year minus the previous year" under the description and "mg/m⁻³/month" under units in the metadata file.
- (3) Save the data and metadata files in non-proprietary software formats such as .txt or .csv files rather than .xlsx files.
- (4) Use well-know and reputable non-proprietary software. Also use software that relies on scripts and modelling rather than a click and point GUI interface.
- (5) Include the actual scripts that you used to analyze your data in the supplementary materials so that a reviewer can rerun them and reproduce your results exactly.

Recommendations #4 and #5 will aid with reproducibility and avoid errors/solutions that can't be replicated. R is the best example of a scripted program that is widely used in ecological research and that will be widely familiar to the PRS-B readership.

My understanding of the analysis is that an identical model, with six latent variables and twenty indicator variables, was fit to each subset of the data grouped according to the 'Methane_seep_site' variable (for 'Mound 12' and 'Jaco Scar') or the 'Activity_category' variable (for 'Active', 'Transitional', and 'Background') after which insignificant paths were pruned stepwise from the SEM.

I downloaded WarpPLS, used 'Data_S3.xlsx' to follow these methods as well as I could with the provided text, but I was unable to replicate the results presented in this manuscript. Perhaps with more detailed information and methods I could have recreated their results, but in the current form, with the current description and the major limitations of WarpPLS this is not possible. Just to be clear: I consider myself an expert SEM user and well-versed in many of the off-the-shelf SEM packages (such as MPLUS and SPSS). All of these canned programs, including WarpPLS, are inferior to an explicitly scripted approach such as R.

I also tried to create the latent variable model in R (using the package lavaan), but also failed because of the small sample size (which ranged from $n = 15$ for 'Mound_12' to $n = 12$ for 'Active' and 'Background') relative to the number of latent variables in the model. The resulting covariance matrix was not positive-definite, which was consistent with the error reported in WarpPLS:

Checking for rank problems ...

The data may be rank deficient, which may lead to misleading results.

The number of data columns is 20.

The number of data rows (usually called the "sample size") is 12.

This problem can often be avoided by having a much larger number of data rows than columns.

Note: WarpPLS did in fact yield results despite inappropriately low sample sizes – just different results than what is reported in the manuscript. Therefore, I am left to believe that some type of non-linear model was fit and that is why our results did not match. Either way, a very general rule of thumb is that a minimum of five replicates is required for each variable in the model; thus, a sample size of ~30 would be required for a model of this complexity. Additional models (i.e., testing among ‘Active’, ‘Transitional’, ‘Background’, etc.) would also require adjustments for familywise error rates associated with multiple comparisons.

Because I could not rebuild the full latent variable model described in Table 1/Figure 2 or recreate the results presented in Figures S3 and S4, I explored the data and relationships using some other statistical approaches, such as latent variable modelling in lavaan for just the relationship between ecosystem function and biodiversity and a principle components analysis (PCA) in place of latent variables.

There are some very influential points in the dataset by virtue of both the small sample size and the huge spread in the data. For example, ‘Faunal_biomass’ within the ‘Active’ subset varies from 0.12 to 571.6 (a ratio of ~ 4763). Consequently, transforming the data may help decrease the influence of individual points in the dataset. I also learned that the PCA showed that the vast majority of the variation (often > 95%) can be explained on the first PC for ‘Biodiversity’, ‘Ecosystem Function’ and ‘Oceanographic Characteristics’. Moreover, the biodiversity ~ ecosystem function relationship was often driven by just a few variables. For example, the correlation between species richness and faunal density in the background sites is 0.86. This suggests that it may be possible to explore some of the properties of the data by focusing on fewer variables and simplifying the analytical approach.

When I plotted the Biodiversity versus Ecosystem Function variables, either as latent variables using the `sem()` command in lavaan or as principle components with `prcomp()`, I produced relationships that look vaguely like those plotted in Figure 3 of the main text, but with significant deviations (my relationships were not significant). I am left to believe that the “standardization” that is produced in Figure 3 is from WarpPLS, and therefore requires an explanation of how to reproduce the relationships that appear in Figure 3.

At this point I feel the need to point out that assigning a direct causal effect of species richness (i.e., Biodiversity) on animal density (i.e., Ecosystem Function) is difficult to justify, as a sampling effect could just as easily lead to higher richness as a function of more organisms per unit area. The same argument goes for several (most?) of the other variables (diversity, evenness, traits, etc.) that are assumed to cause variation in ecosystem function. The application of a causal model in this instance seems poorly justified.

My final point, that is inherently related to my last comment about causality: one of the most revealing aspects of SEM and path analysis is the opportunity to tease apart direct versus indirect effects (the so-called “test of mediation”). In the context of this paper, it would have been very interesting to look at the (for example) direct oceanographic effects on ecosystem function versus the indirect effects that are mediated by biodiversity. Only the indirect effects were analyzed in this study. This raises a severe methodological issue in that it is questionable to draw directional causality from biodiversity to ecosystem functioning in the way that was done so in this paper. The chosen indicators, such as faunal density and species richness, could either effect each other in the opposite way (with more dense communities having higher species richness) or both richness and density could both be controlled by some other mediating variable, such as the many oceanographic indicators. As I pointed out above, those very important indirect models (of which there are many potential indirect effects in this dataset) were not tested in this analysis. Unfortunately, this dataset may be far too small to conduct these critical tests of mediation, and thus I do not think SEM is a good match for these data in the first place.

Despite the authors citing of a paper claiming that PLS and SEM is a “silver bullet” for small sample sizes, I don’t think any reputable biologist or biological statistician would agree that it is acceptable to fit high-dimensional, non-linear latent variable models on a dataset with twelve samples. Therefore, my conclusion (despite the fact that these data are likely among the most difficult and expensive to collect on earth) is that this analysis represents a case of overfitting and one in which the use of non-linear, causal models are neither justified theoretically nor statistically.

Decision letter (RSPB-2020-0828.R0)

26-Oct-2020

Dear Dr Ashford:

I am writing to inform you that your manuscript RSPB-2020-0828 entitled "Biodiversity affects ecosystem functioning differently in chemosynthetic systems invoking new management focus" has, in its current form, been rejected for publication in Proceedings B.

This action has been taken on the advice of referees, who have recommended that substantial revisions are necessary. With this in mind we would be happy to consider a resubmission, provided the comments of the referees are fully addressed. However please note that this is not a provisional acceptance.

- 1) A ‘response to referees’ document including details of how you have responded to the comments, and the adjustments you have made.
- 2) A clean copy of the manuscript and one with 'tracked changes' indicating your 'response to referees' comments document.
- 3) Line numbers in your main document.
- 4) Data - please see our policies on data sharing to ensure that you are complying (<https://royalsociety.org/journals/authors/author-guidelines/#data>).

Sincerely,
Dr Maurine Neiman
<mailto:proceedingsb@royalsociety.org>

Associate Editor
Board Member: 1
Comments to Author:

Thank you for giving Proc B the opportunity to consider this very interesting manuscript. The reviewers had many positive comments and especially appreciated that this MS tests BEF hypotheses in rarely studied, deep-sea, chemosynthetic ecosystems. However, the two statistical reviewers have identified some fundamental issues with the analysis. Reviewer 4 has uncovered some especially concerning questions about the validity of the analytical approach selected. We can only consider a re-submission of this manuscript if all of the concerns raised by reviewers - particularly regarding statistics - are thoroughly addressed.

Reviewer(s)' Comments to Author:

Referee: 1

Comments to the Author(s)

BEF has been rarely studied in deep sea and never studied in chemosynthetic system despite of its importance as a promising model system for BEF research.

It's partly because quantification and robust estimation of ecosystem function is not easy there. This paper indeed did what we needed in BEF study and deep-sea biology in very high quality with multiple robust measures of biodiversity and ecosystem functioning, using samples from Costa Rican cold seeps. The result is very clear and makes sense. It's neat study and I enthusiastically recommend publication after minor revision.

My comments are mostly minor and detailed below. The manuscript is well written and clear, but can be a bit more improved by simplifying the terminology and reducing redundancy (I am sure it can be done easily by the authors).

Line 62-63: The author considers normal deep-sea is still photosynthesis-based, because they rely on POC flux? I think it's good to explain this sort abit for non-deep-sea biologist audience here or somewhere.

Later parts of introduction is already Methods.
Introduction can be shorten in this regard.

'Background' photosynthetically-fuelled samples still have quite low $\delta^{13}\text{C}$ (-21.73) compared with normal sediment. Some comment needed on it?

Font size small in fig 2.

The authors every time say chemosynthetically-fuelled habitat, photosynthetically-fuelled habitat, and the transition zone. You may define in the first appearance and just say seep, non-seep and transitional (or active, background and transition as in Table 2) thereafter wherever details don't matter?

Lines 264-266. Better to combine into one sentence.

Line 285. species diversity means Shannon? May be good to up it in brackets.

The paragraph of Line 327 or somewhere in the discussion: It's good to mention environmental (=oceanographic characteristics) control of biodiversity in non seep system is consistent with our understanding, by citing, eg Sweetman et al 2017 *Elementa*, Yasuhara and Danovaro 2016 *Biological Reviews*.

Line 429 on functional hotspots: Logic is not super clear, because the author's actual result indicate high-diversity non-seep system is more vulnerable from species loss to undermine ecosystem function.

But here in this paragraph the author argue almost opposite "loss of a small number of species could result in the collapse of ecosystem functioning" in seep system.

I think it's good to re-frame the discussion and flow of this paragraph.

Chapman et al 2019. sFDvent: A global trait database for deep-sea hydrothermal-vent fauna. *Global Ecology and Biogeography*: 28, 1538–1551.
This paper is worth mentioning somewhere.

Good to have locality map somewhere.

Moriaki Yasuhara (Signed review)

Referee: 2

Comments to the Author(s)

I would like to thank the authors for producing such an interesting manuscript with such a thorough explanation of key concepts and methodological processes. I really appreciated the opportunity to read and review this manuscript. Here, the authors use deep-sea methane seeps as a natural laboratory, to investigate biodiversity-ecosystem functioning (BEF) relationships along an energy gradient (from high-resource seep to low-resource background communities). Furthermore, they use Structural Equation Modelling to investigate possible environmental drivers influencing the form of BEF relationships in different communities. The aim of the manuscript is, therefore, to test the hypothesis that the form of BEF relationships will change from background, photosynthetically fuelled deep-sea habitats to chemosynthetically fuelled methane seeps, before identifying drivers behind any differences identified. This differs from previous research in two main ways: i) habitat, wherein methane seeps and surrounding seafloor habitats provide a measurable energy gradient to test such a hypothesis and have not been examined in such a way before; and ii) investigating drivers of BEF relationship form beyond topographic characteristics of the seafloor. This manuscript therefore makes an important contribution to the BEF literature, whilst placing methane seeps into a wider ecological literature, making a positive step forward in introducing these sometimes deemed ‘non-traditional’ systems to ecologists familiar with more ‘traditional’ ones as examples of testbeds for ecological theory. I comment on each section of the manuscript below, before making one major comment and more minor ones. I hope that these prove useful.

Title

The title of this manuscript is clear and succinct, explaining the key finding well. I would suggest that the authors consider if there is a way to make it appealing to wider audiences, however, if there are general BEF and energy relationships that might also be present in other ecosystems. I propose this as I found this a very interesting paper and feel it could appeal to a wide range of biologists and ecologists, but might not get ‘picked up’ as often because of the specification of chemosynthetic systems in the title. It appears, based on the discussion, that the management finding might also be applicable in other physiologically demanding systems, so perhaps ‘physiologically challenging systems’ could be a good option.

Abstract

The abstract of this article is clear and concise, providing good insights into the approach and key findings of this work. I wonder whether, instead of the take-home message that ‘photosynthetically-fuelled and chemosynthetically-fuelled environments should be managed differently’, it might also be good to highlight differences in energy/resources which might also apply to other ecosystems.

Introduction

The introduction is very clear. I appreciated the explanations of the different BEF relationship forms under different resource availabilities and this set me up well as a reader to understand your rationale for your hypotheses. I make some suggestions regarding Figure 1 under minor comments, to try to ensure it further complements this well-written text.

Materials and Methods

Section 3.1 of the Materials and Methods section is very detailed and well-explained. I really appreciate this, as I have not sampled seeps, yet could follow the rationale and processes. Section 3.5 is also detailed and clear, with helpful Supporting Information. Contrastingly, I feel section 3.2 could benefit from additional detail beyond that provided in Supporting Information (see Major Comments). For instance, 32 is a rather large number of traits, with impacts on the outcomes of diversity measures if pooled into a multidimensional metric (hence most trait-based studies focusing on ~5 traits). I was also left wondering what the sample by trait matrix was used as an input for, so this needs to be explained. Whilst I have trait-based expertise, I struggled to follow the rationale for this part of the analysis and fear those who aren't familiar with trait-based methods would need even more detail than the extra information I am looking for.

Results

The results are clear and concise and well-supported by further details and re-iteration in the Discussion.

Discussion

I would recommend starting the Discussion section with your main finding(s), rather than re-setting the scene, though I feel the introductory paragraph is strong and the text could be used later in this paragraph and/or moved to the Introduction. I found the Discussion interesting and particularly liked some of the examples used to explain some of the patterns found (e.g. the polychaetes and dominant species). I also found the interpretation regarding a need for investigation in other high resource, physiologically harsh environments compelling and hope that this will fuel future research. Finally, I appreciated the discussion centring on management strategies, as it might not be immediately obvious to readers that there is a clear conservation-oriented message in the findings without this written support.

Supporting Information

The Supporting Information is detailed and complements the work presented in the main manuscript. I appreciated the different forms of the SEMs, as well as the literature review summary, for reference. I make comments on Table S2 under 'Minor Comments' and these relate to the 'Major Comment' below. I really appreciated the provision of trait scoring and references.

Major Comments:

1. My only major comment relates to the use of functional traits in this manuscript, as I find the methods difficult to follow (e.g. I am unclear as to which metrics have been used and why, even after reading the Supporting Information). I appreciated the provision of trait data in the Supporting Information, but struggled to understand how these were processed and used (e.g. it is stated that a suitable matrix is produced but I am unclear as to how this was then used). I was also concerned by the inclusion of 32 traits, which seems to me to be too many (as, if traditional trait-based diversity indices have been used, having so many traits will, essentially, produce values as if every species is unique, thereby almost replicating taxonomic measures). I therefore recommend that more detail is added to the methods (see Minor Comments below) and sensitivity testing is added to Supporting Information, or the trait-based approach is removed from this paper, as I am not clear on how it adds substantially to the testing of the hypothesis set out in the Introduction. I am reluctant to make the latter of these suggestions, as I appreciate the use of a trait-based approach to complement 'traditional' measures of biodiversity, but I am concerned that the approach has not been tested sufficiently for me to be confident in the outcomes of this part of the analysis. It also seems that many of the discussion points centre on specifically measured characteristics and do not depend on the multivariate indices (e.g. lines

330-338 focus in on bioturbation and lines 344-348 highlight an interesting role of dominant species).

Minor Comments:

Line 27: Propose adding '-' after relationships, for consistency with the punctuation in the rest of the sentence.

Line 29: I'm not sure how true this statement is, particularly given the wealth of studies investigating biodiversity and ecosystem functioning relationships in plants. It might be 'safest' to specify 'marine' photosynthesis-based or chemosynthesis-based ecosystems, rather than stating that BEF relationships haven't been explored in detail for photosynthesis-based ecosystems. This is particularly important to consider given the statements in lines 57-60, which suggest that photosynthesis-based ecosystems have been well-studied.

Line 30: I propose adding reference to methane seepage gradients as 'natural laboratories', in line with the main text, as I think this is a great way to highlight deep-sea ecosystems as systems to be considered in a wealth of ecological work.

Line 32: I propose removing 'trait', as 'functional diversity' is more commonly used terminology, or 'trait-based diversity'.

(Lines 38-39: I highlight this sentence as I think it provides a particularly nice summary on the importance of a key finding of this study. I therefore think it would make a good sentence for further communication associated with this work and wanted to 'flag' it as a sentence to 'keep' and re-use (e.g. for research end-user / engagement activities).)

Line 45: I propose changing 'monetarily' to 'economically', as I notice as a reader that this word could easily be mistaken for 'momentarily', if read in a rush, which would be an issue.

Line 46: I propose changing 'extremely important' to 'vital', or another single-word descriptor.

Line 49: I propose changing 'services' to 'contributions' (see comment on line 54 below).

Line 54: A minor comment here that some readers/reviewers might expect the term 'ecosystem services' to be updated given recent IPBES reporting and surrounding discussions terming these 'nature's contributions to people' (e.g. see: <https://www.tandfonline.com/doi/full/10.1080/26395916.2019.1669713#:~:text=In%202017%20the%20term%20was,2017>). It might also be helpful to add a word or two to line 53 to ensure that the link between ecosystem functioning and ecosystem services is clear (e.g. 'in promoting the functioning of ecosystems which, ultimately provide important services to humanity', or an alternative acknowledging the discussions regarding services vs. contributions).

Line 77: I propose changing the word 'expressed', as this might make the reader think of genetic traits, whereas functional traits are characteristics of a species.

Line 83: I wonder if it might be best to just refer to methane seeps (not vents), as this is the focus of this paper and they are a great example on their own (so introducing vents but not including them in your analyses might add unnecessary confusion for readers unfamiliar with chemosynthesis-based ecosystems).

Lines 189-190: I feel that the impact of this choice should be tested for in supplementary methods (e.g. changing the average and testing for the influence of this being 'off' for some species).

Alternatively, there needs to be a justification as to why it is OK to use average individual biomass per sample instead of species-specific values, as this sounds like a limitation, but it is

unclear as to whether it is and, if so, why it is an acceptable one or what could be done to minimise the limitations on the results.

Lines 192-193: 'the opinion of authors' could be summarised as 'expert opinion' (with initials in brackets, as these are helpful for anyone wanting to contact these authors), to ensure that those less familiar with the deep-sea research community understand the 'expert' status of these authors with respect to trait scoring (i.e. I know that these authors are definitely experts, with so much knowledge and experience, but a terrestrial ecologist working on reptiles might not, for example).

Line 211: I propose adding a few words justifying the choice of inverse-distance weighting over other interpolation methods.

Line 283: Trait evenness might be capturing the dominance of certain species in high-resource areas, if evenness increases with resource.

Line 344-345: I just wanted to add a comment that I found similar indications in trait-based studies of hydrothermal-vent ecosystems, so find this particularly interesting as it seems to follow what is also seen at vents.

Lines 355-356: I'm not sure the reference to hydrothermal vents in line 355 is well-placed, as diversity can vary and evenness can be very high at vents, due to the dominance of certain species (contradicting your statement in line 356).

Lines 360-363: Could this also be because the hypothesis, or previous work, is based on an intermediate disturbance hypothesis, whereas seeps have more of a steady gradient (e.g. not a large, abrupt transition like at a vent chimney)?

Lines 408-412: This is an important implication of this research, which I feel could be good to explain a little further in the abstract to capture readers from across disciplines and ecological realms (e.g. the distinction between whole-community conservation and critical species conservation).

Lines 423-426: I feel this is another critical point that should be made earlier in the discussion (e.g. this part could be moved up).

Table 1: I think the metrics 'trait richness, trait diversity, trait evenness' need to be specified in the methods, as well as any package(s) used, as it can influence the results.

Figure 1: This is a helpful figure, complementing the descriptions in the introduction. However, I would recommend adding labels to the lines to further complement the written text, so a reader can instantly see which habitat/hypothesis each line represents (as described in the caption) and some of the key ecological conditions one might expect each type of BEF relationship to occur under (as described in the paragraph starting on line 80).

Figure 2: This is very minor but please consider increasing the font size in this figure, as it is very hard to read.

Figure 3: One data point seems to be driving the shape of plots A and C (e.g. both could be straight without the left-hand points). I therefore feel reference should be made to the supplementary information containing alternative runs, where different options for the setup of the SEM were investigated. Please also consider increasing the font sizes, as this figure is very difficult to read.

Figure S1: I think this is an important supporting figure, but feel the caption would benefit from more detail on what saturating, linear, and accelerating mean, so readers don't need to flick between this and the main manuscript, if possible.

Table S1: This is very helpful. It might be worth checking whether work by Martin Solan, Jasmin Godbold, and colleagues is relevant to cite here (as I understand they have conducted studies into BEF relationships but sometimes found no relationship, which should also be reported on here, if I have remembered this correctly).

Table S2: I'm afraid I'm not sure I understand your description of functional trait richness, as it differs from examples that I have seen, so needs further explanation and justification. This is also the case for 'functional trait diversity'. I have not seen this particular version of this measure in the trait-based literature, so it would be good to know why this approach has been used and how the results compare when using more typically used measures of functional diversity. I am also confused by the description of 'functional trait evenness', as there is a specific functional evenness metric and I'm not sure how this one compares.

Referee: 3

Comments to the Author(s)

This is a very interesting paper that examines biodiversity – ecosystem functioning relationships along a energy input gradient in deep sea communities. My review is primarily focused on the statistical approaches. I can say though, as a terrestrial ecologist, I see direct applicability of the conceptual results to my study systems. I also appreciate the clarity of the writing.

I do have some concerns about the statistical approach. I see none of these as “dealbreaker” issues, but some revision is needed. In some cases, some additional explanation and justification of the approaches taken may be sufficient. In other cases, the authors may want to make revisions to the analyses based on these suggestions.

Most importantly, I was surprised to see that the authors chose to divide their analyses into 5 separate models. I am not sure that this was the right decision for a number of reasons. First, given the relatively low sample sizes, this seems to be cutting these data rather fine. Second, conceptually the authors are treating these sites as a gradient of energy input (Figure 4). Given that they are conceiving these as a gradient, why not analyze as such. Finally, there does not seem to be a compelling practical reason for the separation as the statistical approach can readily handle the nonlinearity.

Sample Size. I recognize that there are substantial logistical limitations to sample sizes in this research area. I am concerned, however, that many of the curvilinear relationships shown in figure 3 are driven by single datapoints. Panels A and C would be linear without the low biodiversity points. That said, I don't think this is a huge problem as the general conclusions from those panels would not be different substantially different minus the low points.

Latent variable selection: The biodiversity latent includes richness, evenness, and diversity measures. Given that evenness and richness measure very different things, are often structured by different ecological factors, and are often uncorrelated, a strong justification is needed to combine them into a single latent variable. For a plant example, see this paper showing how evenness and richness are structured by very different processes in a plant community. Wilsey, B. J., D. R. Chalcraft, et al. (2005). "Relationships among indices suggest that richness is an incomplete surrogate for grassland biodiversity." *Ecology* 86(5): 1178-1184

I would encourage the authors to replace the initial SEM figure in the main text with the fitted figures (S3/4), as those figures are showing the results. Putting dotted lines in to represent NS relationships removed from the models would be a good way of showing the initial SEM form. I

would also encourage some overall fit statistics to be placed on S3/4 when it is moved to the main text.

Overall figure S3/4 could be made much more compact to place all 5 panels together. You may also want to consider ovals to surround latent variables. Many SEM applications (see papers by Grace for examples) by convention use ovals for latent variables and rectangles for directly observed variables.

As a terrestrial plant ecologist, I find figure 4 curious because I normally think of productivity as an ecosystem function. I wonder if characterizing these as high energy input and low energy input systems would be a better approach.

L247. Simply replacing missing values with the mean is a pretty simplistic way of dealing with missing data. Given that strong correlations are likely among environmental values, a more sophisticated approach to estimating these values could easily be used.

Referee: 4

Comments to the Author(s)

I was asked to act as a specialist statistical referee for manuscript RSPB-2020-0828 titled "Biodiversity affects ecosystem functioning differently in chemosynthetic systems invoking new management focus" authored by Ashford et al. I have experience in multi-level modeling and structural equation modelling that makes me a suitable referee for this manuscript. In addition, because a significant focus of my research program is to explore causal relationships between biodiversity and ecosystem function (albeit in very different ecosystems), I also feel qualified to address the appropriateness of statistics and models to answer the paper's core proposed research questions.

The authors provided access to three datasets, but the one pertinent to my review and was used to try and recreate the results was 'Dataset_S3.xlsx'. I appreciate that the authors made their data available, which is key to producing reproducible research. However, there were many mismatches between the variable names in the manuscript and the dataset which hindered reproducibility. These were often simple things that could be fixed with some renaming and reorganization. For example, indicators of the 'Biodiversity' latent variable were referred to as 'trait richness', 'trait diversity', and 'trait evenness', but in the data table they were named 'Functional_richness', 'Functional_diversity', and 'Functional_evenness' (at least the reader is left to assume that these are the same variables). So, regardless of the fate of this manuscript I have several data management and presentation suggestions that will aid in making results reproducible:

(1) Rename the variables in the dataset and to match descriptions in the main text. Use short, meaningful column names rather than variable names such as:

"Seafloor_POC_concentration_for_sampling_year_and_previous_year_mg/m⁻³/month", which is actually the name of one of the columns in Dataset_3.xlsx. A much better choice would have been 'POC'.

(2) Include a metadata file that lists every variable in Dataset_S3.xlsx and provides information about each variable, the units, etc. So, for example, for the 'POC' variable one could then list

"concentration for sampling year minus the previous year" under the description and "mg/m⁻³/month" under units in the metadata file.

(3) Save the data and metadata files in non-proprietary software formats such as .txt or .csv files rather than .xlsx files.

(4) Use well-know and reputable non-proprietary software. Also use software that relies on scripts and modelling rather than a click and point GUI interface.

(5) Include the actual scripts that you used to analyze your data in the supplementary materials so that a reviewer can rerun them and reproduce your results exactly.

Recommendations #4 and #5 will aid with reproducibility and avoid errors/solutions that can't be replicated. R is the best example of a scripted program that is widely used in ecological research and that will be widely familiar to the PRS-B readership.

My understanding of the analysis is that an identical model, with six latent variables and twenty indicator variables, was fit to each subset of the data grouped according to the 'Methane_seep_site' variable (for 'Mound 12' and 'Jaco Scar') or the 'Activity_category' variable (for 'Active', 'Transitional', and 'Background') after which insignificant paths were pruned stepwise from the SEM.

I downloaded WarpPLS, used 'Data_S3.xlsx' to follow these methods as well as I could with the provided text, but I was unable to replicate the results presented in this manuscript. Perhaps with more detailed information and methods I could have recreated their results, but in the current form, with the current description and the major limitations of WarpPLS this is not possible. Just to be clear: I consider myself an expert SEM user and well-versed in many of the off-the-shelf SEM packages (such as MPLUS and SPSS). All of these canned programs, including WarpPLS, are inferior to an explicitly scripted approach such as R.

I also tried to create the latent variable model in R (using the package lavaan), but also failed because of the small sample size (which ranged from $n = 15$ for 'Mound_12' to $n = 12$ for 'Active' and 'Background') relative to the number of latent variables in the model. The resulting covariance matrix was not positive-definite, which was consistent with the error reported in WarpPLS:

Checking for rank problems ...

The data may be rank deficient, which may lead to misleading results.

The number of data columns is 20.

The number of data rows (usually called the "sample size") is 12.

This problem can often be avoided by having a much larger number of data rows than columns.

Note: WarpPLS did in fact yield results despite inappropriately low sample sizes - just different results than what is reported in the manuscript. Therefore, I am left to believe that some type of non-linear model was fit and that is why our results did not match. Either way, a very general rule of thumb is that a minimum of five replicates is required for each variable in the model; thus, a sample size of ~30 would be required for a model of this complexity. Additional models (i.e., testing among 'Active', 'Transitional', 'Background', etc.) would also require adjustments for familywise error rates associated with multiple comparisons.

Because I could not rebuild the full latent variable model described in Table 1/Figure 2 or recreate the results presented in Figures S3 and S4, I explored the data and relationships using some other statistical approaches, such as latent variable modelling in lavaan for just the relationship between ecosystem function and biodiversity and a principle components analysis (PCA) in place of latent variables.

There are some very influential points in the dataset by virtue of both the small sample size and the huge spread in the data. For example, 'Faunal_biomass' within the 'Active' subset varies from 0.12 to 571.6 (a ratio of ~ 4763). Consequently, transforming the data may help decrease the influence of individual points in the dataset. I also learned that the PCA showed that the vast majority of the variation (often > 95%) can be explained on the first PC for 'Biodiversity', 'Ecosystem Function' and 'Oceanographic Characteristics'. Moreover, the biodiversity ~ ecosystem function relationship was often driven by just a few variables. For example, the correlation between species richness and faunal density in the background sites is 0.86. This

suggests that it may be possible to explore some of the properties of the data by focusing on fewer variables and simplifying the analytical approach.

When I plotted the Biodiversity versus Ecosystem Function variables, either as latent variables using the `sem()` command in `lavaan` or as principle components with `prcomp()`, I produced relationships that look vaguely like those plotted in Figure 3 of the main text, but with significant deviations (my relationships were not significant). I am left to believe that the “standardization” that is produced in Figure 3 is from WarpPLS, and therefore requires an explanation of how to reproduce the relationships that appear in Figure 3.

At this point I feel the need to point out that assigning a direct causal effect of species richness (i.e., Biodiversity) on animal density (i.e., Ecosystem Function) is difficult to justify, as a sampling effect could just as easily lead to higher richness as a function of more organisms per unit area. The same argument goes for several (most?) of the other variables (diversity, evenness, traits, etc.) that are assumed to cause variation in ecosystem function. The application of a causal model in this instance seems poorly justified.

My final point, that is inherently related to my last comment about causality: one of the most revealing aspects of SEM and path analysis is the opportunity to tease apart direct versus indirect effects (the so-called “test of mediation”). In the context of this paper, it would have been very interesting to look at the (for example) direct oceanographic effects on ecosystem function versus the indirect effects that are mediated by biodiversity. Only the indirect effects were analyzed in this study. This raises a severe methodological issue in that it is questionable to draw directional causality from biodiversity to ecosystem functioning in the way that was done so in this paper. The chosen indicators, such as faunal density and species richness, could either effect each other in the opposite way (with more dense communities having higher species richness) or both richness and density could both be controlled by some other mediating variable, such as the many oceanographic indicators. As I pointed out above, those very important indirect models (of which there are many potential indirect effects in this dataset) were not tested in this analysis. Unfortunately, this dataset may be far too small to conduct these critical tests of mediation, and thus I do not think SEM is a good match for these data in the first place.

Despite the authors citing of a paper claiming that PLS and SEM is a “silver bullet” for small sample sizes, I don’t think any reputable biologist or biological statistician would agree that it is acceptable to fit high-dimensional, non-linear latent variable models on a dataset with twelve samples. Therefore, my conclusion (despite the fact that these data are likely among the most difficult and expensive to collect on earth) is that this analysis represents a case of overfitting and one in which the use of non-linear, causal models are neither justified theoretically nor statistically.

Author's Response to Decision Letter for (RSPB-2020-0828.R0)

See Appendix A.

RSPB-2021-0950.R0

Review form: Reviewer 3

Recommendation

Major revision is needed (please make suggestions in comments)

It is a condition of publication that authors make their supporting data, code and materials available - either as supplementary material or hosted in an external repository. Please rate, if applicable, the supporting data on the following criteria.

Is it accessible?

No

Is it clear?

No

Is it adequate?

No

Do you have any concerns about statistical analyses in this paper? If so, please specify them explicitly in your report.

Yes

Comments to the Author

This is my second review of this manuscript. I appreciate the efforts that the authors have taken in this revision, and the very detailed response to reviewers.

I should first state that I recognize the complexity of the dataset, the noisy nature of ecological data, and the constraints of limited samples. Given that it is novel to ask these questions about BEF relationships for a chemosynthetic ecosystem, this paper is a valuable contribution regardless of sample size. That said, my feeling is that the authors are taking a statistical sledgehammer to a conceptually straightforward question.

Bottom line, with only 39 samples, I don't think that a model with more than 10 initial predictors (some of which are being modeled using 3 separate smoothers) can be justified.

I recommend that the authors step back and consider their core question – how does diversity influence ecosystem function, and avoid trying to control for every possible variable. The authors lay out a very clear hypothesis in Figure 1, and then go about testing that hypothesis in a very convoluted manner.

From this perspective, the authors have 3 measures of species diversity and 3 measures of functional diversity (or if they choose the reduced variables from the PCA), and two measures of productivity (standing biomass and activity classification). Simple gam models between these variables can be justified a-priori and are likely all that the authors need to answer their core question. This will most likely bring the authors back to the place that they get to in Figure 2 without a whole lot of statistically questionable intermediaries.

A separation of the Species diversity and functional diversity variables may be valuable. Yes, they may be highly correlated, but conceptually they measure very different things. Modeling conceptually different variables separately may help to produce clearer results.

Minor comments:

I appreciate the provision of the R scripts. I had numerous questions about the analytical approach that were unclear in the text but clear in the scripts.

L258-262. It would be very helpful in this paragraph to provide an overview of the analytical approach, particularly to indicate the response and explanatory variable(s) for the gams. As it stands, the following paragraphs require a lot of back and forth because it is not immediately clear what the variables being generated by the PCAs are being used for. I recognize that many of

these details are included in the supplementary information, but it is important to have a broad overview of the analytical approach in the main text.

L277 Again this is confusing because it has not yet been explicitly specified what the explanatory variable(s) in the games are.

Supplement L116 and on. It appears that there are a number of variables that could well be considered random terms here (e.g. cruise). It is not clear to me why these variables even need to be in these models, and if they are included why, they are not included as a random term.

Review form: Reviewer 4

Recommendation

Major revision is needed (please make suggestions in comments)

Scientific importance: Is the manuscript an original and important contribution to its field?

Acceptable

General interest: Is the paper of sufficient general interest?

Marginal

Quality of the paper: Is the overall quality of the paper suitable?

Poor

Is the length of the paper justified?

Yes

Should the paper be seen by a specialist statistical reviewer?

No

Do you have any concerns about statistical analyses in this paper? If so, please specify them explicitly in your report.

No

It is a condition of publication that authors make their supporting data, code and materials available - either as supplementary material or hosted in an external repository. Please rate, if applicable, the supporting data on the following criteria.

Is it accessible?

Yes

Is it clear?

Yes

Is it adequate?

Yes

Do you have any ethical concerns with this paper?

No

Comments to the Author

This is my second review of this manuscript; my first was as a specialist reviewer to assess the analytical and statistical approaches. I mostly limited my previous comments to the SEM model

that was being used to analyze the data, which I found inappropriate based on the small sample size and assumptions of the model. I appreciate that the authors took my comments and suggestions seriously and selected an alternative approach for their analysis. I, like at least on other reviewer of this paper, am a terrestrial plant and animal community ecologist with a history of studying diversity-productivity and diversity-ecosystem function relationships. Consequently, I cannot comment the specific findings or novelty of the paper that relate to benthic marine communities or chemosynthetic habitats, but I can comment on the aspects of the paper that relate to the theory and methods of biodiversity-ecosystem function research.

-- Major comments --

The strength of this manuscript lies in the sampling of rare and novel habitats which create a strong gradient in energy input and, qualitatively, in the source of the energy input (light versus chemosynthetic). Even though I don't know this ecosystem or the deep-sea literature, I do feel like the gradient of habitats sampled and the data collected represent a potentially important contribution to the and improve our understanding of how this unique ecosystem works. However, I feel that casting this as a paper that can teach us something general and broad about the biodiversity - ecosystem function (BEF) relationship is misguided. The small sample size, lack of experimental manipulations, static nature of the data collection (i.e., no time-series data), and lack of true measures of ecosystem services make for a poor fit to answer BEF types of questions. There have literally been hundreds of BEF papers and some of them have used high dimensional models to analyze thousands of samples from experiments distributed across the globe. So, while I think the gradient of energy or resources creates an interesting template across which to explore patterns of richness, density, biomass and species traits I think the results and conclusions go too far in attempting to shed light on general BEF theory in what amounts to a very unique habitat.

One criticism which I think needs to be addressed is how this manuscript treats and defines ecosystem services in the first place. In all ecosystem function literature which I have read (which amounts to quite a lot over the years) the quantifiable "services" are typically processes that are measured by rates. Typical measures include variables like NPP, rates of biogeochemical cycling, flood control, etc., all which can be measured as an amount of something that is produced, cycled or mitigated over time. But here, the authors use at least three ecosystem functions that to me are perplexing: faunal density, faunal biomass and individual body size. The other two metrics, which include bioturbation potential and calcification metric, appear to be much more in line with what I expect are typical ecosystem services. But the other three are very problematic because, while they may be correlated with rates of energy flow or biochemical transformations (although not necessarily), they are not surrogates for ecosystem services and in fact they are confounded in important ways.

Body size should be thought of as a species trait that is shaped by evolution and life history. Indeed, your citation list supports that this is a species trait rather than an ecosystem services. For example, the paper by Chapman et al. "sFDvent: A global trait database for deep-sea hydrothermal-vent fauna" listed body size as a key species trait that can influence ecosystem function, not a measure of ecosystem function itself.

With respect to faunal density, I attempted to raise this issue in my previous review but perhaps it was missed because of the many other comments: the cause and effect of richness and faunal density appear to be confounded in this manuscript. Faunal density is being measured as an ecosystem service (i.e., as a response variable) that is causally related to species richness and diversity, whereas the reverse causal relationship is equally likely to be true. In plant communities it has often been observed that as the number of individuals in a patch increases so too does the number of species because of the sampling effect. At high density strong competitive effects then reduce the number of species in a patch. So the question becomes what biotic or abiotic factors are controlling the density of species in communities, an issue raised by the authors themselves in lines 83 - 88. Ecologists have long discussed these issues and several well-established methods, such as the well-known 2001 paper by Gotelli and Colwell in *Ecology*

Letters, which describes sample- and individual-based rarefaction and accumulation curves to compare richness while correcting for differences in density. But even more problematic for the current manuscript is the fact that the authors acknowledge that energy or resource inputs should control species density but then they use species density as an ecosystem service (i.e. a response variable) that putatively responds to diversity. The result is that the variables are highly confounded, begging the question of: what variable is causally controlling what other variables?

With respect to biomass, the problem is similar to density: what is the factor that ultimately controls the biomass, is it resources or diversity? As is rightfully acknowledged in the introduction, including lines 96-101, resources influence biomass accumulation which controls species richness. In the BEF literature that is cited throughout the manuscript (i.e., Tilman, Naeem, Hector, Hooper, etc.) the central question was, after controlling for the negative effect of biomass on diversity, how much of a residual effect of diversity is there on the biomass and primary production? The highly correlated and confounded nature of density, diversity and function are what drove ecologists to tease apart these relationships with well-planned highly controlled experiments such as the European BIODEPTH experiment, Cedar Creek LTER and nutnet.

I don't doubt the bivariate shape of the relationships described in the results section of the manuscript, but I do doubt that one can assign any causal explanation for what gives rise to the shape of these relationships without more data and a better model.

So, in the end my main suggestion is to reframe the paper as an observational exploration of how these bivariate relationships change across gradients of energy and resource inputs and tone down the discussion of BEF, which as I have expressed, I think is poorly addressed by the data and experimental design. Alternatively one could focus truly on services that are not confounded with density and maintain the hypotheses based on BEF. The authors are in a better position to know if bioturbation and biogenic calcium carbonate content are clear representations of ecosystem services, but density and body size must be eliminated as focal ecosystem services. More appropriate is to analyze density and body size as stand-alone response variables (which then answers a very different ecological question) or as covariates in an analysis focused on other response variables.

If the authors choose to continue to frame their hypotheses in terms of ecosystem function, there are some potentially important missing references from this paper that may help to clarify their questions. These include the 1994 "green book" edited by ED Schulze and HA Mooney entitled "Biodiversity and Ecosystem Function" (and references therein) and a 1996 book (and references therein) edited by Mooney, Cushman, Medina, Sala and Schulze titled "Functional roles of biodiversity: a global perspective." The idea that ecosystem function is influenced by species loss or gain, and that the nature of this relationship is influenced by which species are lost from or added to a system, was described by Vitousek and Hooper in the green book and in a model by Sala et al. in the 1996 Mooney et al. book. It is my feeling that the authors of this paper need to read some of this classic BEF literature to provide appropriate context for their ideas and hypotheses. I know that their newer citations are based on these more classic texts, but I was surprised none showed up in the reference list.

-- Minor comments --

Line 79-81 and Figure 1: see my note above about missing literature, references and ideas previously described in the BEF literature.

Line 72-73: This seems to be a reference about deep sea ecosystem. If so, please clarify and add more detailed information about the findings of citation 19.

Line 90: I don't agree that resource and energy gradients are synonymous as assumed here, but it would be interesting to hear why the authors thought this was so. This line deserves a bit more

information and perhaps some citations. I recommend a book edited by RE Ricklefs and D Schluter titled "Species Diversity in Ecological Communities" and (references therein) as well as richness-energy papers published by David Currie in Am Nat. To me energy and resources differ based on scale. Energy enters ecosystems via production but it is not something species compete for per se. Resources on the other hand are being competed for and display heterogeneity over smaller scales than energy.

Lines 246-249: You mentioned primary productivity of the sites and listed Table 1 and Table S4 but I never again saw mention of this important variable and I did not see it presented in either of the tables. I also failed to find it in dataset S3. This seems like an important covariate or predictor that should be part of the analysis but I think it was omitted.

Lines 298-317: Reporting the correlations with PCA factors is important, but I found this section lengthy and not to be composed of particularly interesting ecological results. This is almost more of a technical result that could be greatly shortened and largely presented in the supplemental materials. As I discussed at length above, the creation of PCA variables that are of such differing and contrasting nature is problematic and should be reconsidered.

Decision letter (RSPB-2021-0950.R0)

11-Jun-2021

Dear Dr Ashford:

Your manuscript has now been peer reviewed and the reviews have been assessed by an Associate Editor. The reviewers' comments (not including confidential comments to the Editor) and the comments from the Associate Editor are included at the end of this email for your reference. As you will see, the reviewers and the Editors have raised some concerns with your manuscript and we would like to invite you to revise your manuscript to address them.

Research ethics:

Use of animals and field studies:

It is a condition of publication that you make available the data and research materials supporting the results in the article (<https://royalsociety.org/journals/authors/author-guidelines/#data>). Datasets should be deposited in an appropriate publicly available repository and details of the associated accession number, link or DOI to the datasets must be included in the Data Accessibility section of the article (<https://royalsociety.org/journals/ethics-policies/data-sharing-mining/>). Reference(s) to datasets should also be included in the reference list of the article with DOIs (where available).

Please submit a copy of your revised paper within three weeks. If we do not hear from you within this time your manuscript will be rejected. If you are unable to meet this deadline please let us know as soon as possible, as we may be able to grant a short extension.

Best wishes,
 Dr Maurine Neiman
 mailto: proceedingsb@royalsociety.org

Associate Editor

Comments to Author:

Thank you for your careful attention to the reviewers' comments in the last revision. Two reviewers have now provided an additional round of commentary. They have highlighted some important points:

1. REFRAMING: Reviewer 4 points out that some of the metrics used in this study are potentially confounded. In brief, "ecosystem services" are usually rates (e.g., NPP). Static variables (e.g., faunal density) could be correlated with rates, or they might not be. Also note that the direction of causality is in question here (does high diversity cause high faunal density or does high faunal density cause high diversity through sampling effects?). Reviewer 4 suggests that you either reframe the paper as an observational exploration of how your bivariate relationships change across gradients of energy and resource inputs OR focus exclusively on true "ecosystem services" (i.e., bioturbation and biogenic calcium, if these are indeed useful indicators in this ecosystem). Whichever path you choose, note that the value of this paper lies in (1) the gradient in energy, (2) the source of resource input (chemosynthesis), and (3) the fact that it took place in a novel habitat. Due to low replication, lack of manipulation, lack of time series data, and lack of "true" measures of ecosystem services, it has less to teach us about how BEF relationships work generally than other (better replicated, manipulated, tracked over time, measuring true ecosystem services) studies might; instead of reaching for broad conclusions relevant to BEF in other ecosystems, the paper should instead play to its strengths.

2. STATISTICAL MODELS: The move from SEMs to GAMs has greatly improved the MS. However, Reviewer 3 is suggesting some adjustments to the GAMs. I agree that these adjustments are needed, as the number of initial predictors (10) is inappropriate for the sample size (39). Separating out the models into multiple, simpler models will produce a more straightforward, easy-to-interpret analysis. Several fixed effects currently included in the GAMs might be more appropriately modeled as random effects. And the MS needs to make the model formulations crystal clear. Speaking of the methods...

3. METHODS: I recognize that there are some detailed methods that are more appropriately relegated to a reference or the supplement, but a reader should be able to get a rough idea of what you've done without resorting to reading references or supplement. Make sure that all methods are at least briefly sketched (e.g., bioturbation) in the Methods section, even if the nitty-gritty details can only be provided in a reference. This goes especially for the statistical models. See this reference for some excellent suggestions on how to report model formulation: <https://besjournals.onlinelibrary.wiley.com/doi/full/10.1111/2041-210X.12577>

These changes and the rest suggested by the reviewers below must be made before we can consider a revised version of the MS.

Reviewer(s)' Comments to Author:

Referee: 3

Comments to the Author(s).

This is my second review of this manuscript. I appreciate the efforts that the authors have taken in this revision, and the very detailed response to reviewers.

I should first state that I recognize the complexity of the dataset, the noisy nature of ecological data, and the constraints of limited samples. Given that it is novel to ask these questions about BEF relationships for a chemosynthetic ecosystem, this paper is a valuable contribution regardless of sample size. That said, my feeling is that the authors are taking a statistical sledgehammer to a conceptually straightforward question.

Bottom line, with only 39 samples, I don't think that a model with more than 10 initial predictors (some of which are being modeled using 3 separate smoothers) can be justified.

I recommend that the authors step back and consider their core question – how does diversity influence ecosystem function, and avoid trying to control for every possible variable. The authors lay out a very clear hypothesis in Figure 1, and then go about testing that hypothesis in a very convoluted manner.

From this perspective, the authors have 3 measures of species diversity and 3 measures of functional diversity (or if they choose the reduced variables from the PCA), and two measures of productivity (standing biomass and activity classification). Simple gam models between these variables can be justified a-priori and are likely all that the authors need to answer their core question. This will most likely bring the authors back to the place that they get to in Figure 2 without a whole lot of statistically questionable intermediaries.

A separation of the Species diversity and functional diversity variables may be valuable. Yes, they may be highly correlated, but conceptually they measure very different things. Modeling conceptually different variables separately may help to produce clearer results.

Minor comments:

I appreciate the provision of the R scripts. I had numerous questions about the analytical approach that were unclear in the text but clear in the scripts.

L258-262. It would be very helpful in this paragraph to provide an overview of the analytical approach, particularly to indicate the response and explanatory variable(s) for the gams. As it stands, the following paragraphs require a lot of back and forth because it is not immediately clear what the variables being generated by the PCAs are being used for. I recognize that many of these details are included in the supplementary information, but it is important to have a broad overview of the analytical approach in the main text.

L277 Again this is confusing because it has not yet been explicitly specified what the explanatory variable(s) in the gams are.

Supplement L116 and on. It appears that there are a number of variables that could well be considered random terms here (e.g. cruise). It is not clear to me why these variables even need to be in these models, and if they are included why, they are not included as a random term.

Referee: 4

Comments to the Author(s).

This is my second review of this manuscript; my first was as a specialist reviewer to assess the analytical and statistical approaches. I mostly limited my previous comments to the SEM model that was being used to analyze the data, which I found inappropriate based on the small sample size and assumptions of the model. I appreciate that the authors took my comments and suggestions seriously and selected an alternative approach for their analysis. I, like at least on other reviewer of this paper, am a terrestrial plant and animal community ecologist with a history of studying diversity-productivity and diversity-ecosystem function relationships. Consequently, I cannot comment the specific findings or novelty of the paper that relate to benthic marine communities or chemosynthetic habitats, but I can comment on the aspects of the paper that relate to the theory and methods of biodiversity-ecosystem function research.

-- Major comments --

The strength of this manuscript lies in the sampling of rare and novel habitats which create a strong gradient in energy input and, qualitatively, in the source of the energy input (light versus chemosynthetic). Even though I don't know this ecosystem or the deep-sea literature, I do feel

like the gradient of habitats sampled and the data collected represent a potentially important contribution to the and improve our understanding of how this unique ecosystem works.

However, I feel that casting this as a paper that can teach us something general and broad about the biodiversity - ecosystem function (BEF) relationship is misguided. The small sample size, lack of experimental manipulations, static nature of the data collection (i.e., no time-series data), and lack of true measures of ecosystem services make for a poor fit to answer BEF types of questions. There have literally been hundreds of BEF papers and some of them have used high dimensional models to analyze thousands of samples from experiments distributed across the globe. So, while I think the gradient of energy or resources creates an interesting template across which to explore patterns of richness, density, biomass and species traits I think the results and conclusions go too far in attempting to shed light on general BEF theory in what amounts to a very unique habitat.

One criticism which I think needs to be addressed is how this manuscript treats and defines ecosystem services in the first place. In all ecosystem function literature which I have read (which amounts to quite a lot over the years) the quantifiable "services" are typically processes that are measured by rates. Typical measures include variables like NPP, rates of biogeochemical cycling, flood control, etc., all which can be measured as an amount of something that is produced, cycled or mitigated over time. But here, the authors use at least three ecosystem functions that to me are perplexing: faunal density, faunal biomass and individual body size. The other two metrics, which include bioturbation potential and calcification metric, appear to be much more in line with what I expect are typical ecosystem services. But the other three are very problematic because, while they may be correlated with rates of energy flow or biochemical transformations (although not necessarily), they are not surrogates for ecosystem services and in fact they are confounded in important ways.

Body size should be thought of as a species trait that is shaped by evolution and life history. Indeed, your citation list supports that this is a species trait rather than an ecosystem services. For example, the paper by Chapman et al. "sFDvent: A global trait database for deep-sea hydrothermal-vent fauna" listed body size as a key species trait that can influence ecosystem function, not a measure of ecosystem function itself.

With respect to faunal density, I attempted to raise this issue in my previous review but perhaps it was missed because of the many other comments: the cause and effect of richness and faunal density appear to be confounded in this manuscript. Faunal density is being measured as an ecosystem service (i.e., as a response variable) that is causally related to species richness and diversity, whereas the reverse causal relationship is equally likely to be true. In plant communities it has often been observed that as the number of individuals in a patch increases so too does the number of species because of the sampling effect. At high density strong competitive effects then reduce the number of species in a patch. So the question becomes what biotic or abiotic factors are controlling the density of species in communities, an issue raised by the authors themselves in lines 83 - 88. Ecologists have long discussed these issues and several well-established methods, such as the well-known 2001 paper by Gotelli and Colwell in *Ecology Letters*, which describes sample- and individual-based rarefaction and accumulation curves to compare richness while correcting for differences in density. But even more problematic for the current manuscript is the fact that the authors acknowledge that energy or resource inputs should control species density but then they use species density as an ecosystem service (i.e. a response variable) that putatively responds to diversity. The result is that the variables are highly confounded, begging the question of: what variable is causally controlling what other variables?

With respect to biomass, the problem is similar to density: what is the factor that ultimately controls the biomass, is it resources or diversity? As is rightfully acknowledged in the introduction, including lines 96-101, resources influence biomass accumulation which controls species richness. In the BEF literature that is cited throughout the manuscript (i.e., Tilman, Naem, Hector, Hooper, etc.) the central question was, after controlling for the negative effect of biomass on diversity, how much of a residual effect of diversity is there on the biomass and primary production? The highly correlated and confounded nature of density, diversity and

function are what drove ecologists to tease apart these relationships with well-planned highly controlled experiments such as the European BIODEPTH experiment, Cedar Creek LTER and nutnet.

I don't doubt the bivariate shape of the relationships described in the results section of the manuscript, but I do doubt that one can assign any causal explanation for what gives rise to the shape of these relationships without more data and a better model.

So, in the end my main suggestion is to reframe the paper as an observational exploration of how these bivariate relationships change across gradients of energy and resource inputs and tone down the discussion of BEF, which as I have expressed, I think is poorly addressed by the data and experimental design. Alternatively one could focus truly on services that are not confounded with density and maintain the hypotheses based on BEF. The authors are in a better position to know if bioturbation and biogenic calcium carbonate content are clear representations of ecosystem services, but density and body size must be eliminated as focal ecosystem services. More appropriate is to analyze density and body size as stand-alone response variables (which then answers a very different ecological question) or as covariates in an analysis focused on other response variables.

If the authors choose to continue to frame their hypotheses in terms of ecosystem function, there are some potentially important missing references from this paper that may help to clarify their questions. These include the 1994 "green book" edited by ED Schulze and HA Mooney entitled "Biodiversity and Ecosystem Function" (and references therein) and a 1996 book (and references therein) edited by Mooney, Cushman, Medina, Sala and Schulze titled "Functional roles of biodiversity: a global perspective." The idea that ecosystem function is influenced by species loss or gain, and that the nature of this relationship is influenced by which species are lost from or added to a system, was described by Vitousek and Hooper in the green book and in a model by Sala et al. in the 1996 Mooney et al. book. It is my feeling that the authors of this paper need to read some of this classic BEF literature to provide appropriate context for their ideas and hypotheses. I know that their newer citations are based on these more classic texts, but I was surprised none showed up in the reference list.

-- Minor comments --

Line 79-81 and Figure 1: see my note above about missing literature, references and ideas previously described in the BEF literature.

Line 72-73: This seems to be a reference about deep sea ecosystem. If so, please clarify and add more detailed information about the findings of citation 19.

Line 90: I don't agree that resource and energy gradients are synonymous as assumed here, but it would be interesting to hear why the authors thought this was so. This line deserves a bit more information and perhaps some citations. I recommend a book edited by RE Ricklefs and D Schluter titled "Species Diversity in Ecological Communities" and (references therein) as well as richness-energy papers published by David Currie in Am Nat. To me energy and resources differ based on scale. Energy enters ecosystems via production but it is not something species compete for per se. Resources on the other hand are being competed for and display heterogeneity over smaller scales than energy.

Lines 246-249: You mentioned primary productivity of the sites and listed Table 1 and Table S4 but I never again saw mention of this important variable and I did not see it presented in either of the tables. I also failed to find it in dataset S3. This seems like an important covariate or predictor that should be part of the analysis but I think it was omitted.

Lines 298-317: Reporting the correlations with PCA factors is important, but I found this section lengthy and not to be composed of particularly interesting ecological results. This is almost more

of a technical result that could be greatly shortened and largely presented in the supplemental materials. As I discussed at length above, the creation of PCA variables that are of such differing and contrasting nature is problematic and should be reconsidered.

Author's Response to Decision Letter for (RSPB-2021-0950.R0)

See Appendix B.

Decision letter (RSPB-2021-0950.R1)

23-Jul-2021

Dear Dr Ashford

I am pleased to inform you that your manuscript entitled "Relationships between biodiversity and ecosystem functioning proxies strengthen when approaching chemosynthetic deep-sea methane seeps" has been accepted for publication in Proceedings B.

Data Accessibility section

Open Access

Your article has been estimated as being 9 pages long. Our Production Office will be able to confirm the exact length at proof stage.

Paper charges

Sincerely,
Dr Maurine Neiman
Editor, Proceedings B
mailto: proceedingsb@royalsociety.org

Associate Editor:

Comments to Author:

The authors have done an extremely thorough job of integrating the reviewers' extensive suggestions into this version of the MS. I am now happy to accept this paper for publication in Proc B.

Appendix A

Response to Referee Comments

‘Biodiversity affects ecosystem functioning differently in chemosynthetic systems invoking new management focus’

RSPB-2020-0828

Associate Editor

Comments to the Author:

Thank you for giving Proc B the opportunity to consider this very interesting manuscript. The reviewers had many positive comments and especially appreciated that this MS tests BEF hypotheses in rarely studied, deep-sea, chemosynthetic ecosystems. However, the two statistical reviewers have identified some fundamental issues with the analysis. Reviewer 4 has uncovered some especially concerning questions about the validity of the analytical approach selected. We can only consider a re-submission of this manuscript if all of the concerns raised by reviewers - particularly regarding statistics - are thoroughly addressed.

The authors thank Moriaki Yasuhara and the three anonymous referees and Associate Editor for generously providing their valuable time to review this manuscript. The comments offered are insightful and constructive, and we believe that the manuscript has been strengthened as a result of the review process. Please see below for a point-by-point response by the authors (text highlighted in red) to these comments. Important edits are emphasised using italics. Note: all references made by the authors to line numbers reflect those in the revised manuscript with tracked changes accepted (file name: ‘Ashford et al. manuscript tracked’).

Reviewers' Comments to Authors:

Referee: 1

BEF has been rarely studied in deep sea and never studied in chemosynthetic system despite of its importance as a promising model system for BEF research.

It's partly because quantification and robust estimation of ecosystem function is not easy there.

This paper indeed did what we needed in BEF study and deep-sea biology in very high quality with multiple robust measures of biodiversity and ecosystem functioning, using samples from Costa Rican cold seeps. The result is very clear and makes sense. It's neat study and I enthusiastically recommend publication after minor revision.

My comments are mostly minor and detailed below. The manuscript is well written and clear, but can be a bit more improved by simplifying the terminology and reducing redundancy (I am sure it can be done easily by the authors).

Line 62-63: The author considers normal deep-sea is still photosynthesis-based, because they rely on POC flux? I think it's good to explain this sort a bit for non-deep-sea biologist audience here or somewhere.

We agree that this would be helpful for non-deep-sea biologists, so have changed lines 110 – 114 to “Here we utilise deep-sea methane seeps and the surrounding seafloor as a natural laboratory to investigate how the form of BEF relationships change along gradients of resource supply; moving from *typical lower-resource deep-sea habitats primarily dependent upon the delivery of photosynthetically-fixed carbon from surface waters*, to higher-resource chemosynthetically-fuelled habitats”.

Later parts of introduction is already Methods.
Introduction can be shorten in this regard.

We agree, and so have and have moved text from lines 136 - 143 into the methods section.

‘Background’ photosynthetically-fuelled samples still have quite low $\delta^{13}\text{C}$ (-21.73) compared with normal sediment. Some comment needed on it?

We believe that an organic $\delta^{13}\text{C}$ of -21 ‰ or -22 ‰ is typical of continental margin sediments beneath oxygen minimum zones, and of the fauna that consume these sediments. For example, see figure 3 of Cowie, G.L., Mowbray, S., Lewis, M., Matheson, H., McKenzie, R., 2009. Carbon and nitrogen elemental and stable isotopic compositions of surficial sediments from the Pakistan margin of the Arabian Sea. *Deep-Sea Research II* 56, 271–282. For the East Pacific, in Levin and Michener 2002 (*Limnology and Oceanography* 47: 1336-1345), the photosynthetically fuelled non seep infaunal signatures had average $\delta^{13}\text{C}$ values of -21.39‰, -20.78‰ and -20.53‰ at seeps in the Gulf of Alaska, Oregon margin, and Eel River margin, respectively.

Font size small in fig 2.

As part of the statistical revisions to the manuscript, we have removed the original Figure 2 from the manuscript.

The authors every time say chemosynthetically-fuelled habitat, photosynthetically-fuelled habitat, and the transition zone. You may define in the first appearance and just say seep, non-seep and transitional (or active, background and transition as in Table 2) thereafter wherever details don’t matter?

We agree that there is some redundancy in our repetition of ‘chemosynthetically-fuelled’ and ‘photosynthetically-fuelled’ (now revised to ‘chemosynthetically-dependent’ and ‘photosynthetically-dependent’). However, since *Proceedings B* is not a specialist deep-sea journal, and many readers may not be deep-sea scientists, we feel this repetition is helpful to remind readers of the carbon source of each type of habitat.

Lines 264–266. Better to combine into one sentence.

This sentence has now been removed as a result of the re-wording of the results section following the revision of statistical methods.

Line 285. species diversity means Shannon? May be good to up it in brackets.

Yes, this is correct – the Shannon Index was used as a metric of species diversity. We have clarified this in Section 3.2, Table 1, and Table S2.

The paragraph of Line 327 or somewhere in the discussion: It’s good to mention environmental (=oceanographic characteristics) control of biodiversity in non seep system is consistent with our understanding, by citing, eg Sweetman et al 2017 *Elementa*, Yasuhara and Danovaro 2016 *Biological Reviews*.

We are in agreement with this. However, with the revision of statistical methodology in this paper, the section relating to relationships between environment and biodiversity has been removed.

Line 429 on functional hotspots: Logic is not super clear, because the author’s actual result indicate high-diversity non-seep system is more vulnerable from species loss to undermine ecosystem function.

But here in this paragraph the author argue almost opposite “loss of a small number of species could result in the collapse of ecosystem functioning” in seep system.

I think it’s good to re-frame the discussion and flow of this paragraph.

We agree that this section requires clarification. In the case of seep systems, the loss of a small number of high functioning species could result in the collapse of ecosystem functioning. This section now reads: *“The lower alpha diversity of resource-rich chemosynthetically-dependent environments (42, 69-71) coupled with the importance of a specialised selection of species in driving ecosystem functioning imply that the loss of a small number of these critical species could result in a major decline in ecosystem functioning”*.

Chapman et al 2019. sFDvent: A global trait database for deep- sea hydrothermal- vent fauna. *Global Ecology and Biogeography*: 28, 1538–1551.

This paper is worth mentioning somewhere.

We agree that this paper is an important contribution to scientific understanding of traits in chemosynthetic ecosystems. However, because of the focus of the paper on vent fauna, we do not feel that there is a natural place to cite this paper within the current manuscript.

Good to have locality map somewhere.

We agree and have added a locality map as a new Figure S2 (Supplementary Materials).

Moriaki Yasuhara (Signed review)

Referee: 2

I would like to thank the authors for producing such an interesting manuscript with such a thorough explanation of key concepts and methodological processes. I really appreciated the opportunity to read and review this manuscript. Here, the authors use deep-sea methane seeps as a natural laboratory, to investigate biodiversity-ecosystem functioning (BEF) relationships along an energy gradient (from high-resource seep to low-resource background communities). Furthermore, they use Structural Equation Modelling to investigate possible environmental drivers influencing the form of BEF relationships in different communities. The aim of the manuscript is, therefore, to test the hypothesis that the form of BEF relationships will change from background, photosynthetically fuelled deep-sea habitats to chemosynthetically fuelled methane seeps, before identifying drivers behind any differences identified. This differs from previous research in two main ways: i) habitat, wherein methane seeps and surrounding seafloor habitats provide a measurable energy gradient to test such a hypothesis and have not been examined in such a way before; and ii) investigating drivers of BEF relationship form beyond topographic characteristics of the seafloor. This manuscript therefore makes an important contribution to the BEF literature, whilst placing methane seeps into a wider ecological literature, making a positive step forward in introducing these sometimes deemed ‘non-traditional’ systems to ecologists familiar with more ‘traditional’ ones as examples of testbeds for ecological theory. I comment on each section of the manuscript below, before making one major comment and more minor ones. I hope that these prove useful.

Title:

The title of this manuscript is clear and succinct, explaining the key finding well. I would suggest that the authors consider if there is a way to make it appealing to wider audiences, however, if there are general BEF and energy relationships that might also be present in other ecosystems. I propose this as I found this a very interesting paper and feel it could appeal to a wide range of biologists and ecologists, but might not get ‘picked up’ as often because of the specification of chemosynthetic systems in the title. It appears, based on the discussion, that the management finding might also be applicable in other physiologically demanding systems, so perhaps ‘physiologically challenging systems’ could be a good option.

We agree that a change of title may increase readership amongst non-deep-sea biologist audiences. We have changed the title to ‘Biodiversity affects ecosystem functioning differently in a physiologically demanding chemosynthetic ecosystem, invoking new management focus’.

Abstract:

The abstract of this article is clear and concise, providing good insights into the approach and key findings of this work. I wonder whether, instead of the take-home message that ‘photosynthetically-fuelled and chemosynthetically-fuelled environments should be managed differently’, it might also be good to highlight differences in energy/resources which might also apply to other ecosystems.

We believe that the finding that “photosynthetically-dependent and chemosynthetically-dependent environments should be managed differently to maintain peak ecosystem functioning under the threat of human disturbance” is the major focus of this manuscript, and should remain as the take-home message of the abstract. We agree that it would be good to highlight that differences in energy between other ecosystems may also influence BEF relationships, and promote further investigation of BEF relationships in non-traditional resource environments in the abstract (lines 48-49).

Introduction:

The introduction is very clear. I appreciated the explanations of the different BEF relationship forms under different resource availabilities and this set me up well as a reader to understand your rationale for your hypotheses. I make some suggestions regarding Figure 1 under minor comments, to try to ensure it further complements this well-written text.

Materials and Methods:

Section 3.1 of the Materials and Methods section is very detailed and well-explained. I really appreciate this, as I have not sampled seeps, yet could follow the rationale and processes. Section 3.5 is also detailed and clear, with helpful Supporting Information. Contrastingly, I feel section 3.2 could benefit from additional detail beyond that provided in Supporting Information (see Major Comments). For instance, 32 is a rather large number of traits, with impacts on the outcomes of diversity measures if pooled into a multidimensional metric (hence most trait-based studies focusing on ~5 traits). I was also left wondering what the sample by trait matrix was used as an input for, so this needs to be explained. Whilst I have trait-based expertise, I struggled to follow the rationale for this part of the analysis and fear those who aren’t familiar with trait-based methods would need even more detail than the extra information I am looking for.

We agree that the original text provided did not provide sufficient detail. We have now revised this section to the following: “Functional *trait* biodiversity was quantified based on the scoring of taxa *at family level* for 32 functional traits in 7 groupings (Dataset S1). Maximum affiliation with a trait was scored a number ‘5’, whilst no affiliation with a trait was scored a number ‘0’ – please see Ashford *et al.* (45) for complete detail on the creation of this dataset. For each sample, the taxon-specific scores for functional traits (0 – 5) were multiplied by the abundance of that taxon and totalled across all taxa present, producing a sample (rows) by functional trait abundance (columns) matrix. Functional *trait richness, diversity and evenness* were calculated from this functional trait matrix using the same methodology as if the matrix were of sample by species abundance type”. Please also see comments below for further discussion of the approach taken.

Results:

The results are clear and concise and well-supported by further details and re-iteration in the Discussion.

Discussion:

I would recommend starting the Discussion section with your main finding(s), rather than re-setting the scene, though I feel the introductory paragraph is strong and the text could be used later in this paragraph and/or moved to the Introduction. I found the Discussion interesting and particularly liked some of the examples used to explain some of the patterns found (e.g. the polychaetes and dominant species). I also found the interpretation regarding a need for investigation in other high resource,

physiologically harsh environments compelling and hope that this will fuel future research. Finally, I appreciated the discussion centring on management strategies, as it might not be immediately obvious to readers that there is a clear conservation-oriented message in the findings without this written support.

Thank you for your feedback. With regards to the opening of the discussion section, before going into major findings, we would prefer to use the first paragraph to remind readers of the aim of the paper. That way, any readers who are skimming the paper (for instance, just reading abstract and discussion), will be aware of the overall aims of the paper.

Supporting Information:

The Supporting Information is detailed and complements the work presented in the main manuscript. I appreciated the different forms of the SEMs, as well as the literature review summary, for reference. I make comments on Table S2 under ‘Minor Comments’ and these relate to the ‘Major Comment’ below. I really appreciated the provision of trait scoring and references.

Major Comments:

1. My only major comment relates to the use of functional traits in this manuscript, as I find the methods difficult to follow (e.g. I am unclear as to which metrics have been used and why, even after reading the Supporting Information). I appreciated the provision of trait data in the Supporting Information, but struggled to understand how these were processed and used (e.g. it is stated that a suitable matrix is produced but I am unclear as to how this was then used). I was also concerned by the inclusion of 32 traits, which seems to me to be too many (as, if traditional trait-based diversity indices have been used, having so many traits will, essentially, produce values as if every species is unique, thereby almost replicating taxonomic measures). I therefore recommend that more detail is added to the methods (see Minor Comments below) and sensitivity testing is added to Supporting Information, or the trait-based approach is removed from this paper, as I am not clear on how it adds substantially to the testing of the hypothesis set out in the Introduction. I am reluctant to make the latter of these suggestions, as I appreciate the use of a trait-based approach to complement ‘traditional’ measures of biodiversity, but I am concerned that the approach has not been tested sufficiently for me to be confident in the outcomes of this part of the analysis. It also seems that many of the discussion points centre on specifically measured characteristics and do not depend on the multivariate indices (e.g. lines 330-338 focus in on bioturbation and lines 344-348 highlight an interesting role of dominant species).

As outlined above, we agree that the original text did not provide sufficient detail. Please see our response above (Materials and Methods section). The approach taken is not that of traditional trait-based diversity indices. However, neither is it completely new, being similar to the approach taken when using Rao’s quadratic entropy to measure functional diversity – for example, see Botta-Dukát, Z. 2005. ‘Rao's quadratic entropy as a measure of functional diversity based on multiple traits’. *Journal of Vegetation Science* 16: 533:540; and similar to the approach taken by Alfaro-Lucas et al. 2020. ‘High environmental stress and productivity increase functional diversity along a deep-sea hydrothermal vent gradient’. *Ecology* 101: e03144.

With regards to the value added by using functional trait diversity alongside taxonomic diversity measures, this method, being based on family-level trait data provides a complimentary approach, giving a fuller picture of biodiversity in the study system.

Minor Comments:

Line 27: Propose adding ‘-’ after relationships, for consistency with the punctuation in the rest of the sentence.

We agree, and have made this change.

Line 29: I’m not sure how true this statement is, particularly given the wealth of studies investigating

biodiversity and ecosystem functioning relationships in plants. It might be 'safest' to specify 'marine' photosynthesis-based or chemosynthesis-based ecosystems, rather than stating that BEF relationships haven't been explored in detail for photosynthesis-based ecosystems. This is particularly important to consider given the statements in lines 57-60, which suggest that photosynthesis-based ecosystems have been well-studied.

We agree, and have made this change.

Line 30: I propose adding reference to methane seepage gradients as 'natural laboratories', in line with the main text, as I think this is a great way to highlight deep-sea ecosystems as systems to be considered in a wealth of ecological work.

We agree, and have made this change.

Line 32: I propose removing 'trait', as 'functional diversity' is more commonly used terminology, or 'trait-based diversity'.

On the request of Reviewer 4, we elected to use the term 'functional trait diversity' throughout the manuscript.

(Lines 38-39: I highlight this sentence as I think it provides a particularly nice summary on the importance of a key finding of this study. I therefore think it would make a good sentence for further communication associated with this work and wanted to 'flag' it as a sentence to 'keep' and re-use (e.g. for research end-user / engagement activities).)

Thank you for your advice – we have marked the sentence for use in any further communication associated with the work.

Line 45: I propose changing 'monetarily' to 'economically', as I notice as a reader that this word could easily be mistaken for 'momentarily', if read in a rush, which would be an issue.

We agree, and have made this change.

Line 46: I propose changing 'extremely important' to 'vital', or another single-word descriptor.

We agree that a single-word descriptor would be preferable, and have changed 'extremely important' to 'fundamental'.

Line 49: I propose changing 'services' to 'contributions' (see comment on line 54 below).

We agree, and have made this change.

Line 54: A minor comment here that some readers/reviewers might expect the term 'ecosystem services' to be updated given recent IPBES reporting and surrounding discussions terming these 'nature's contributions to people' (e.g.

see: <https://www.tandfonline.com/doi/full/10.1080/26395916.2019.1669713#:~:text=In%202017%20the%20term%20was,2017>). It might also be helpful to add a word or two to line 53 to ensure that the link between ecosystem functioning and ecosystem services is clear (e.g. 'in promoting the functioning of ecosystems which, ultimately provide important services to humanity', or an alternative acknowledging the discussions regarding services vs. contributions).

We agree. This sentence now reads "...scientists have been working to understand the role of biodiversity in promoting *and maintaining* the functioning of ecosystems, *and therefore, the provision of ecosystem services to humanity...*".

Line 77: I propose changing the word 'expressed', as this might make the reader think of genetic traits, whereas functional traits are characteristics of a species.

We agree, and have changed the word to 'present'.

Line 83: I wonder if it might be best to just refer to methane seeps (not vents), as this is the focus of this paper and they are a great example on their own (so introducing vents but not including them in your analyses might add unnecessary confusion for readers unfamiliar with chemosynthesis-based ecosystems).

We agree, and have made this change.

Lines 189-190: I feel that the impact of this choice should be tested for in supplementary methods (e.g. changing the average and testing for the influence of this being ‘off’ for some species). Alternatively, there needs to be a justification as to why it is OK to use average individual biomass per sample instead of species-specific values, as this sounds like a limitation, but it is unclear as to whether it is and, if so, why it is an acceptable one or what could be done to minimise the limitations on the results.

We agree that use of average individual body size is a limitation of this study. Use of this metric was born from the high observed species richness, low observed abundance of many of these species, and use of wet-weight measuring; it was not possible to accurately measure the average body weight for each species individually (as now detailed in the supplementary methods). Wet weight was used because the alternative – dry weight or carbon weight, would be a destructive process, and we wanted to keep specimens for future use by others in the Scripps Institution of Oceanography Benthic Invertebrate Collection (some specimens were new species, for example). Because of the principal component analysis approach now taken in this revised manuscript, where average body size is only one of five functioning proxies forming each PC axis, we do not believe that any potential limitations of this approach will significantly impact our overall results.

Lines 192-193: ‘the opinion of authors’ could be summarised as ‘expert opinion’ (with initials in brackets, as these are helpful for anyone wanting to contact these authors), to ensure that those less familiar with the deep-sea research community understand the ‘expert’ status of these authors with respect to trait scoring (i.e. I know that these authors are definitely experts, with so much knowledge and experience, but a terrestrial ecologist working on reptiles might not, for example).

We agree, and have made this change.

Line 211: I propose adding a few words justifying the choice of inverse-distance weighting over other interpolation methods.

We have added the following sentence to the supplementary methods at line 72: “Inverse Distance Weighting (IDW) was used because it provides a good compromise between accuracy of outputs and computational demand (Maleika, 2020)”.

Reference: Maleika, W. 2020. ‘Inverse distance weighting method optimization in the process of digital terrain model creation based on data collected from a multibeam echosounder’. Applied Geomatics 4: 397-407.

Line 283: Trait evenness might be capturing the dominance of certain species in high-resource areas, if evenness increases with resource.

We agree with this point. However, with the revisions made to the statistical methods of this paper, this section has now been removed.

Line 344-345: I just wanted to add a comment that I found similar indications in trait-based studies of hydrothermal-vent ecosystems, so find this particularly interesting as it seems to follow what is also seen at vents.

This is very interesting, thank you. If you could point me towards your paper, I would be interested in reading through it.

Lines 355-356: I’m not sure the reference to hydrothermal vents in line 355 is well-placed, as

diversity can vary and evenness can be very high at vents, due to the dominance of certain species (contradicting your statement in line 356).

We have changed the wording here from ‘often’ to ‘may be’.

Lines 360-363: Could this also be because the hypothesis, or previous work, is based on an intermediate disturbance hypothesis, whereas seeps have more of a steady gradient (e.g. not a large, abrupt transition like at a vent chimney)?

We agree that the ‘chemotone’ transitional area at seeps is not likely to be as abrupt as at vents, and that this may facilitate this. We have made changes to wording to reflect this: “*Made possible by the breadth of the chemotone at seeps, BEF relationships in this zone may reflect a dynamic interaction of opposing BEF relationships between the unimodal form characteristic of active chemosynthetically-dependent habitat (driven by physiological constraints, seep specialism and competition), and the typically accelerating form characteristic of background photosynthetically-dependent deep-sea habitat (driven by positive species interactions)*”.

Lines 408-412: This is an important implication of this research, which I feel could be good to explain a little further in the abstract to capture readers from across disciplines and ecological realms (e.g. the distinction between whole-community conservation and critical species conservation).

We agree and added mention to this in the abstract between lines 38-41.

Lines 423-426: I feel this is another critical point that should be made earlier in the discussion (e.g. this part could be moved up).

We feel that this point can only be made with sufficient impact following prior paragraphs, and so prefer to leave it as is.

Table 1: I think the metrics ‘trait richness, trait diversity, trait evenness’ need to be specified in the methods, as well as any package(s) used, as it can influence the results.

We agree, and have made changes to the Materials and Methods of the manuscript (please see comments above).

Figure 1: This is a helpful figure, complementing the descriptions in the introduction. However, I would recommend adding labels to the lines to further complement the written text, so a reader can instantly see which habitat/hypothesis each line represents (as described in the caption) and some of the key ecological conditions one might expect each type of BEF relationship to occur under (as described in the paragraph starting on line 80).

We agree and have substantially revised this figure:

Figure 2: This is very minor but please consider increasing the font size in this figure, as it is very hard to read.

With the revisions made to statistical methods, this figure has now been removed.

Figure 3: One data point seems to be driving the shape of plots A and C (e.g. both could be straight without the left-hand points). I therefore feel reference should be made to the supplementary information containing alternative runs, where different options for the setup of the SEM were investigated. Please also consider increasing the font sizes, as this figure is very difficult to read.

With the revisions made to statistical methods, this figure has now been removed.

Figure S1: I think this is an important supporting figure, but feel the caption would benefit from more detail on what saturating, linear, and accelerating mean, so readers don't need to flick between this and the main manuscript, if possible.

We have now added labels to the lines on this figure to make it easier to comprehend, and have revised the caption, which now reads "Conceptual diagram illustrating how the rate of change in ecosystem functioning with biodiversity loss is influenced by the form of biodiversity – ecosystem functioning (BEF) relationship. Type of BEF relationship: red dotted line – saturating (*ecosystem functioning increases with biodiversity in an asymptotic manner*); orange dashed line – linear (*ecosystem functioning increases with biodiversity in a linear manner*); black dotted and dashed line – accelerating (*ecosystem functioning increases with biodiversity in an exponential manner*)".

Table S1: This is very helpful. It might be worth checking whether work by Martin Solan, Jasmin Godbold, and colleagues is relevant to cite here (as I understand they have conducted studies into BEF relationships but sometimes found no relationship, which should also be reported on here, if I have remembered this correctly).

We agree that the work of Martin Solan and Jasmin Godbold is highly relevant here, and should feature in this table. The following paper has been included as an additional row to the table: Godbold, J.A., Solan, M. 2009. 'Relative importance of biodiversity and the abiotic environment in mediating an ecosystem process'. *Marine Ecology Progress Series* 396: 273-282.

Table S2: I'm afraid I'm not sure I understand your description of functional trait richness, as it

differs from examples that I have seen, so needs further explanation and justification. This is also the case for ‘functional trait diversity’. I have not seen this particular version of this measure in the trait-based literature, so it would be good to know why this approach has been used and how the results compare when using more typically used measures of functional diversity. I am also confused by the description of ‘functional trait evenness’, as there is a specific functional evenness metric and I’m not sure how this one compares.

We have revised the Materials and Methods section of the manuscript to clarify the approach taken here. Please see our responses above for further discussion on this topic.

Referee: 3

This is a very interesting paper that examines biodiversity – ecosystem functioning relationships along an energy input gradient in deep-sea communities. My review is primarily focused on the statistical approaches. I can say though, as a terrestrial ecologist, I see direct applicability of the conceptual results to my study systems. I also appreciate the clarity of the writing.

I do have some concerns about the statistical approach. I see none of these as “dealbreaker” issues, but some revision is needed. In some cases, some additional explanation and justification of the approaches taken may be sufficient. In other cases, the authors may want to make revisions to the analyses based on these suggestions.

Most importantly, I was surprised to see that the authors chose to divide their analyses into 5 separate models. I am not sure that this was the right decision for a number of reasons. First, given the relatively low sample sizes, this seems to be cutting these data rather fine. Second, conceptually the authors are treating these sites as a gradient of energy input (Figure 4). Given that they are conceiving these as a gradient, why not analyze as such. Finally, there does not seem to be a compelling practical reason for the separation as the statistical approach can readily handle the nonlinearity.

We agree, and have made revisions to our statistical approach. As now detailed in a revised section 3.5 of the manuscript (and in greater detail in the supplementary methods), we have:

- Standardised variables to place them on equivalent scales.
- Reduced dimensionality using Principal Component Analysis.
- Investigated relationships between biodiversity and ecosystem functioning using Generalised Additive Models (GAMs). All habitat types (background/ transition/ active seep) are analysed concurrently in the same model, with an interaction term between habitat type and biodiversity making it possible to fit a distinct line for BEF relationships for each habitat type, within this broader model.

Sample Size: I recognize that there are substantial logistical limitations to sample sizes in this research area. I am concerned, however, that many of the curvilinear relationships shown in figure 3 are driven by single datapoints. Panels A and C would be linear without the low biodiversity points. That said, I don’t think this is a huge problem as the general conclusions from those panels would not be different substantially different minus the low points.

Thank you for recognising the logistical obstacles to obtaining large sample sizes in deep-sea ecosystems. We believe that the new analysis approach helps to mitigate some of the drawbacks of the relatively low sample size.

Latent variable selection: The biodiversity latent includes richness, evenness, and diversity measures. Given that evenness and richness measure very different things, are often structured by different ecological factors, and are often uncorrelated, a strong justification is needed to combine them into a single latent variable. For a plant example, see this paper showing how evenness and

richness are structured by very different processes in a plant community. Wilsey, B. J., D. R. Chalcraft, et al. (2005). "Relationships among indices suggest that richness is an incomplete surrogate for grassland biodiversity." *Ecology* 86(5): 1178-1184

With our revised statistical approach, we have now used Principal Component Analysis to reduce dimensionality, as opposed to defining latent variables. The weighting of biodiversity metrics on the first two principal components (which together are able to explain 83.37 % of variance) is given in Table 3. From this, one can see that all biodiversity metrics, perhaps save 'functional trait evenness', show a very similar weighting on PC1. Biodiversity variables show a more variable weighting on PC2 (Table 3), with more of a division between richness and evenness variables evident. These differences between PC1 and PC2 allow for relationships between biodiversity and ecosystem functioning to be teased apart in more detail than previously, where a single latent variable was used to represent biodiversity. For example, see lines 430-449.

I would encourage the authors to replace the initial SEM figure in the main text with the fitted figures (S3/4), as those figures are showing the results. Putting dotted lines in to represent NS relationships removed from the models would be a good way of showing the initial SEM form. I would also encourage some overall fit statistics to be placed on S3/4 when it is moved to the main text.

Thank you for the suggestions, but with the change in statistical approach, these figures have now been removed from the manuscript.

Overall figure S3/4 could be made much more compact to place all 5 panels together. You may also want to consider ovals to surround latent variables. Many SEM applications (see papers by Grace for examples) by convention use ovals for latent variables and rectangles for directly observed variables.

Thank you for these suggestions, but, as noted above, with the change in statistical approach, these figures have now been removed from the manuscript.

As a terrestrial plant ecologist, I find figure 4 curious because I normally think of productivity as an ecosystem function. I wonder if characterizing these as high energy input and low energy input systems would be a better approach.

We agree, and have clarified this terminology in the paper. For example, in the abstract (line 31), line 90 (the first mention of resource supply in the introduction), and line 422 (the first mention of resource supply in the discussion).

L247. Simply replacing missing values with the mean is a pretty simplistic way of dealing with missing data. Given that strong correlations are likely among environmental values, a more sophisticated approach to estimating these values could easily be used.

Missing environmental data represents only 5 of 342 observations (1.5%) in this study. We agree that alternative ways of dealing with missing data may theoretically be more appropriate. However, in practice, we do not believe that using a more sophisticated method would significantly alter the results of our analyses, especially considering the subsequent combination of variables into PC axes.

Referee: 4

I was asked to act as a specialist statistical referee for manuscript RSPB-2020-0828 titled "Biodiversity affects ecosystem functioning differently in chemosynthetic systems invoking new management focus" authored by Ashford et al. I have experience in multi-level modeling and structural equation modelling that makes me a suitable referee for this manuscript. In addition, because a significant focus of my research program is to explore causal relationships between biodiversity and ecosystem function (albeit in very different ecosystems), I also feel qualified to address the appropriateness of statistics and models to answer the paper's core proposed research questions.

The authors provided access to three datasets, but the one pertinent to my review and was used to try and recreate the results was 'Dataset_S3.xlsx'. I appreciate that the authors made their data available, which is key to producing reproducible research. However, there were many mismatches between the variable names in the manuscript and the dataset which hindered reproducibility. These were often simple things that could be fixed with some renaming and reorganization. For example, indicators of the 'Biodiversity' latent variable were referred to as 'trait richness', 'trait diversity', and 'trait evenness', but in the data table they were named 'Functional_richness', 'Functional_diversity', and 'Functional_evenness' (at least the reader is left to assume that these are the same variables). So, regardless of the fate of this manuscript I have several data management and presentation suggestions that will aid in making results reproducible:

(1) Rename the variables in the dataset and to match descriptions in the main text. Use short, meaningful column names rather than variable names such as: "Seafloor_POC_concentration_for_sampling_year_and_previous_year_mg/m⁻³/month", which is actually the name of one of the columns in Dataset_3.xlsx. A much better choice would have been 'POC'.

We agree with the value of this, and have revised all variable names in Dataset S3 to ensure they match those used in the manuscript, and are as concise as possible without losing meaning.

(2) Include a metadata file that lists every variable in Dataset_S3.xlsx and provides information about each variable, the units, etc. So, for example, for the 'POC' variable one could then list "concentration for sampling year minus the previous year" under the description and "mg/m⁻³/month" under units in the metadata file.

We agree with the value of this, and have included this information in Tables S2-S4 of the 'Supplementary_Figures_and_Tables' file.

(3) Save the data and metadata files in non-proprietary software formats such as .txt or .csv files rather than .xlsx files.

We agree, and have now provided Dataset_S3 as a .csv file.

(4) Use well-know and reputable non-proprietary software. Also use software that relies on scripts and modelling rather than a click and point GUI interface.

As part of our reanalysis of data, we have now switched to using R software.

(5) Include the actual scripts that you used to analyze your data in the supplementary materials so that a reviewer can rerun them and reproduce your results exactly.

Actual scripts have now been provided in the Supplementary materials – please see file 'R analysis script.txt'.

Recommendations #4 and #5 will aid with reproducibility and avoid errors/solutions that can't be replicated. R is the best example of a scripted program that is widely used in ecological research and that will be widely familiar to the PRS-B readership.

My understanding of the analysis is that an identical model, with six latent variables and twenty indicator variables, was fit to each subset of the data grouped according to the 'Methane_seep_site' variable (for 'Mound 12' and 'Jaco Scar') or the 'Activity_category' variable (for 'Active', 'Transitional', and 'Background') after which insignificant paths were pruned stepwise from the SEM.

I downloaded WarpPLS, used 'Data_S3.xlsx' to follow these methods as well as I could with the

provided text, but I was unable to replicate the results presented in this manuscript. Perhaps with more detailed information and methods I could have recreated their results, but in the current form, with the current description and the major limitations of WarpPLS this is not possible. Just to be clear: I consider myself an expert SEM user and well-versed in many of the off-the-shelf SEM packages (such as MPLUS and SPSS). All of these canned programs, including WarpPLS, are inferior to an explicitly scripted approach such as R.

I also tried to create the latent variable model in R (using the package lavaan), but also failed because of the small sample size (which ranged from $n = 15$ for 'Mound_12' to $n = 12$ for 'Active' and 'Background') relative to the number of latent variables in the model. The resulting covariance matrix was not positive-definite, which was consistent with the error reported in WarpPLS:

Checking for rank problems ...

The data may be rank deficient, which may lead to misleading results.

The number of data columns is 20.

The number of data rows (usually called the "sample size") is 12.

This problem can often be avoided by having a much larger number of data rows than columns.

Note: WarpPLS did in fact yield results despite inappropriately low sample sizes – just different results than what is reported in the manuscript. Therefore, I am left to believe that some type of non-linear model was fit and that is why our results did not match. Either way, a very general rule of thumb is that a minimum of five replicates is required for each variable in the model; thus, a sample size of ~30 would be required for a model of this complexity. Additional models (i.e., testing among 'Active', 'Transitional', 'Background', etc.) would also require adjustments for familywise error rates associated with multiple comparisons.

Because I could not rebuild the full latent variable model described in Table 1/Figure 2 or recreate the results presented in Figures S3 and S4, I explored the data and relationships using some other statistical approaches, such as latent variable modelling in lavaan for just the relationship between ecosystem function and biodiversity and a principle components analysis (PCA) in place of latent variables.

There are some very influential points in the dataset by virtue of both the small sample size and the huge spread in the data. For example, 'Faunal_biomass' within the 'Active' subset varies from 0.12 to 571.6 (a ratio of ~ 4763). Consequently, transforming the data may help decrease the influence of individual points in the dataset. I also learned that the PCA showed that the vast majority of the variation (often > 95%) can be explained on the first PC for 'Biodiversity', 'Ecosystem Function' and 'Oceanographic Characteristics'. Moreover, the biodiversity ~ ecosystem function relationship was often driven by just a few variables. For example, the correlation between species richness and faunal density in the background sites is 0.86. This suggests that it may be possible to explore some of the properties of the data by focusing on fewer variables and simplifying the analytical approach.

When I plotted the Biodiversity versus Ecosystem Function variables, either as latent variables using the sem() command in lavaan or as principle components with prcomp(), I produced relationships that look vaguely like those plotted in Figure 3 of the main text, but with significant deviations (my relationships were not significant). I am left to believe that the "standardization" that is produced in Figure 3 is from WarpPLS, and therefore requires an explanation of how to reproduce the relationships that appear in Figure 3.

At this point I feel the need to point out that assigning a direct causal effect of species richness (i.e., Biodiversity) on animal density (i.e., Ecosystem Function) is difficult to justify, as a sampling effect

could just as easily lead to higher richness as a function of more organisms per unit area. The same argument goes for several (most?) of the other variables (diversity, evenness, traits, etc.) that are assumed to cause variation in ecosystem function. The application of a causal model in this instance seems poorly justified.

Thank you for this detailed investigation of the statistical approach taken. Based on your comments, we have made major revisions to our statistical approach. This is now detailed in a revised section 3.5 of the manuscript and covered in full in the supplementary methods (see below). To summarise, we have:

- Standardised variables to place them on equivalent scales (mean of 0, standard deviation of 1).
- Reduced dimensionality in the dataset using Principal Component Analysis.
- Investigated relationships between biodiversity and ecosystem functioning using Generalised Additive Models (GAMs), which do not assume causality.

As the reviewer highlights, often the vast majority of variation can be explained on the first principal component for ‘ecosystem functioning’, ‘biodiversity’, ‘oceanographic’ and ‘terrain’ variables (please see Table 1 for definitions of these groupings, Table 2 for details of variance explained by PC1 and PC2, and Table 3 for weightings of individual metrics on PC axes). Using this approach, we were able to characterise the six biodiversity variables as two principal components, the five ecosystem functioning variables as two principal components, the five oceanographic variables as two principal components, and characterise seafloor slope and seafloor ruggedness together on a single principal component.

We used a GAM, which is not a causal model, to facilitate the modelling of potentially non-linear BEF relationships whilst taking into account the influence of other variables. We analysed all habitat types (background/ transition/ active seep) concurrently in the same model. We used an interaction term between habitat type and biodiversity to make it possible to fit a distinct BEF relationship for each habitat type, within this broader model.

Given the limitations of the original analysis approach highlighted by the reviewer, we believe that this new analysis approach increases the confidence that can be placed in our findings.

Revised section 3.5 of the manuscript:

“The form of BEF relationships at active chemosynthetically-dependent, transitional ‘chemotone’, and background photosynthetically-dependent habitats was investigated via a Generalised Additive Model (GAM) framework in R v3.4.2 using the package ‘mgcv 1.8-33’ (47, 51). Prior to analysis, continuous variables were standardised in R v3.4.2 to ensure equivalence of scales. Missing environmental data (5 values of 342 observations) were replaced with ‘0’ after standardisation. Dimensionality in the dataset was reduced by undertaking Principal Component Analysis (PCA) in R v3.4.2 using the function ‘pr.comp’. For ecosystem functioning proxies, the first two principal components captured 89.8 % of variance, whilst for biodiversity variables, the first two principal components captured 83.4 % of variance (Table 1, Table 2). For oceanographic variables, the first two principal components captured 99.5 % of variance; and for terrain variables, the first principal component captured 99.5 % of variance (Table 1, Table 2). See Table 3 for details of variable weighting on each PC axis.

Smoothers were specified for all continuous independent variables, and optimised automatically using the Generalised Cross Validation criterion (51). A different smooth was specified between biodiversity and ecosystem functioning proxies for each sample activity category, in accordance with our hypothesis that the form of BEF relationship is habitat dependent. Model performance diagnostics and the Akaike Information Criterion (AIC) were used to select appropriate error distributions and

link functions (52). GAMs were refined from the initial model via backward stepwise selection by consideration of independent variable p -values and model AIC. The number of knots specified for each smoothed term was optimised based on model AIC values and assumptions (see Table S5 for final model structures). Please see Supplementary Materials for additional methodological detail”.

Full detail provided in Supplementary Methods:

“1.4 Statistical Analyses

Because we hypothesised relationships between biodiversity and ecosystem functioning would be non-linear, the form of BEF relationships at active chemosynthetically-dependent, transitional ‘chemotone’, and background photosynthetically-dependent habitats was investigated via a Generalised Additive Model (GAM) framework in R v3.4.2 using the package ‘mgcv 1.8-33’ (5, 6). Prior to analysis, sample ‘AD4919, PC4 + PC8’ (Quepos Landslide, ‘Active’) was removed from the dataset because it contained no macrofauna, and continuous variables were standardised in R v3.4.2 (mean of zero, standard deviation of ± 1) to ensure equivalence of scales. Missing environmental data (5 values of 342 observations) were replaced with ‘0’ after standardisation. Because dimensionality in the dataset was high relative to the degree of replication, and because covariance was high within some variable groupings, we reduced dimensionality by undertaking Principal Component Analysis (PCA) in R v3.4.2 using the function ‘pr.comp’. For ecosystem functioning proxies, the first two principal components captured 89.8 % of variance, whilst for biodiversity variables, the first two principal components captured 83.4 % of variance (Table 1, Table 2). For oceanographic variables, the first two principal components captured 99.5 % of variance; and for terrain variables, the first principal component captured 99.5 % of variance (Table 1, Table 2). See Table 3 for details of variable weighting on each PC axis.

With ecosystem functioning proxies PC1 or PC2 as the dependent variable, initial GAMs included as independent variables PC1 and PC2 of the biodiversity variables PCA and oceanographic variables PCA, PC1 of the terrain variables PCA, and the variables ‘Terrain Position Index’, ‘ δ ORP’, ‘activity category’, ‘cruise identifier’, and ‘methane seep site’. Smoothers were specified for all continuous variables, with a penalised thinplate regression spline used as the smoothing function, and smoothing parameters optimised automatically using the Generalised Cross Validation criterion (6). A different smooth was specified between biodiversity and ecosystem functioning proxies for each sample activity category, in accordance with our hypothesis that the form of BEF relationship is habitat dependent. Model performance diagnostics and the Akaike Information Criterion (AIC) were used to select appropriate error distributions and link functions (7), and acceptable satisfaction of model assumptions was investigated using the ‘gam.check’ function (6). GAMs were refined from the initial model via backward stepwise selection by consideration of independent variable p -values and model AIC until a minimum AIC value was reached. The number of knots specified for each smoothed term was optimised based on model AIC values and assumptions, with care taken to avoid overfitting (see Table S5 for final model structures). Models were visualised using the package ‘visreg 2.7.0’ (8). An R script is provided within the Supplementary Materials, detailing the analysis process”.

My final point, that is inherently related to my last comment about causality: one of the most revealing aspects of SEM and path analysis is the opportunity to tease apart direct versus indirect effects (the so-called “test of mediation”). In the context of this paper, it would have been very interesting to look at the (for example) direct oceanographic effects on ecosystem function versus the indirect effects that are mediated by biodiversity. Only the indirect effects were analyzed in this study. This raises a severe methodological issue in that it is questionable to draw directional causality from biodiversity to ecosystem functioning in the way that was done so in this paper. The chosen indicators, such as faunal density and species richness, could either effect each other in the opposite way (with more dense communities having higher species richness) or both richness and density could

both be controlled by some other mediating variable, such as the many oceanographic indicators. As I pointed out above, those very important indirect models (of which there are many potential indirect effects in this dataset) were not tested in this analysis. Unfortunately, this dataset may be far too small to conduct these critical tests of mediation, and thus I do not think SEM is a good match for these data in the first place.

We are in agreement, and believe that the new analysis approach taken is more appropriate, considering the limitations of the data and the hypotheses being tested.

Despite the authors citing of a paper claiming that PLS and SEM is a “silver bullet” for small sample sizes, I don’t think any reputable biologist or biological statistician would agree that it is acceptable to fit high-dimensional, non-linear latent variable models on a dataset with twelve samples. Therefore, my conclusion (despite the fact that these data are likely among the most difficult and expensive to collect on earth) is that this analysis represents a case of overfitting and one in which the use of non-linear, causal models are neither justified theoretically nor statistically.

Thank you for your comments on the statistical aspects of this manuscript. We believe that the revisions made as a result of them have increased the confidence that can be placed in our conclusions.

Appendix B

Response to Review Comments

‘Relationships between biodiversity and ecosystem functioning proxies strengthen when approaching chemosynthetic deep-sea methane seeps’

**Original title: ‘Biodiversity affects ecosystem functioning differently in a physiologically-stressful chemosynthetic ecosystem, invoking new management focus’
RSPB-2021-0950**

The authors thank the two anonymous referees and Associate Editor for generously providing their valuable time to review this manuscript. The comments offered are insightful and constructive, and we believe that the manuscript has been strengthened as a result of the review process. Please see below for a point-by-point response by the authors (text highlighted in red font) to these comments. Important edits to text are emphasised using italics. Note: all references made by the authors to line numbers reflect those in the revised manuscript with tracked changes (file name: ‘Ashford et al. manuscript tracked’).

Associate Editor

Comments to the Author:

Thank you for your careful attention to the reviewers' comments in the last revision. Two reviewers have now provided an additional round of commentary. They have highlighted some important points:

1. REFRAMING: Reviewer 4 points out that some of the metrics used in this study are potentially confounded. In brief, "ecosystem services" are usually rates (e.g., NPP). Static variables (e.g., faunal density) could be correlated with rates, or they might not be. Also note that the direction of causality is in question here (does high diversity cause high faunal density or does high faunal density cause high diversity through sampling effects?). Reviewer 4 suggests that you either reframe the paper as an observational exploration of how your bivariate relationships change across gradients of energy and resource inputs OR focus exclusively on true "ecosystem services" (i.e., bioturbation and biogenic calcium, if these are indeed useful indicators in this ecosystem). Whichever path you choose, note that the value of this paper lies in (1) the gradient in energy, (2) the source of resource input (chemosynthesis), and (3) the fact that it took place in a novel habitat. Due to low replication, lack of manipulation, lack of time series data, and lack of "true" measures of ecosystem services, it has less to teach us about how BEF relationships work generally than other (better replicated, manipulated, tracked over time, measuring true ecosystem services) studies might; instead of reaching for broad conclusions relevant to BEF in other ecosystems, the paper should instead play to its strengths.

Thank you for these useful comments. In response, we have decided to reframe the manuscript as an mensurative study, using a natural gradient from typical photosynthetically-fuelled deep-sea environments to chemosynthetically-fuelled methane seep habitats to address key questions regarding the role of resource gradients in shaping correlations between biodiversity and proxies for ecosystem functioning. In brief, we have:

- Emphasised that our unique mensurative study uses natural gradients to investigate *correlations* between variables only, as opposed to investigating relationships between biodiversity and ecosystem functioning in a traditional experimental way. We have removed all mention or allusion to causality in the relationships investigated, and revised wording accordingly.
- Reduced the degree to which traditional BEF relationships are discussed in the Introduction and Discussion.
- Revised text to avoid any statements that may mislead readers into thinking that our results can be directly extrapolated to other marine or terrestrial environments.

- Removed text making broad conclusions or suggestions regarding BEF relationships based on the results we obtained.
- Removed body size from models as a proxy for ecosystem functioning.

Please see our responses to the comments of Reviewer 4 below for much greater detail on these points.

2. STATISTICAL MODELS: The move from SEMs to GAMs has greatly improved the MS. However, Reviewer 3 is suggesting some adjustments to the GAMs. I agree that these adjustments are needed, as the number of initial predictors (10) is inappropriate for the sample size (39). Separating out the models into multiple, simpler models will produce a more straightforward, easy-to-interpret analysis. Several fixed effects currently included in the GAMs might be more appropriately modelled as random effects. And the MS needs to make the model formulations crystal clear. Speaking of the methods...

We very much appreciate the additional statistical advice, and have made changes to our models as a result:

- We have separated out the taxonomic and functional trait aspects of biodiversity, and modelled them independently.
- We investigated modelling some variables as random effects (cruise identity and methane seep site), but concluded that the models could be simplified further without loss of power by removing the variables in question entirely (suggested as an option by Reviewer 3).

The resulting models describe very similar relationships between variables as in our original models, but reveal these relationships in a less convoluted manner. Model formulation has been further clarified in the manuscript, with dependent and independent variables clearly labelled in Section 3.5, initial model structure specified in the main text manuscript (lines 257 - 261), and detailed final model structures given as Table S6. Please see our response to Reviewer 3 for greater detail on these points.

3. METHODS: I recognize that there are some detailed methods that are more appropriately relegated to a reference or the supplement, but a reader should be able to get a rough idea of what you've done without resorting to reading references or supplement. Make sure that all methods are at least briefly sketched (e.g., bioturbation) in the Methods section, even if the nitty-gritty details can only be provided in a reference. This goes especially for the statistical models. See this reference for some excellent suggestions on how to report model formulation:

<https://besjournals.onlinelibrary.wiley.com/doi/full/10.1111/2041-210X.12577>

Thank you for these suggestions. We have added further detail back into the study methods, with particular attention paid to the estimation of bioturbation activity (please see revised section 3.3 – lines 189 - 198). Following the suggestion of Zuur and Ieno (2016) – the reference you kindly suggested above – we have now specified initial GAM model structure in the main text of the manuscript (lines 257 - 261), and have detailed final GAM model structures as Table S6.

These changes and the rest suggested by the reviewers below must be made before we can consider a revised version of the MS.

Reviewers' Comments to Authors:

Referee: 3

This is my second review of this manuscript. I appreciate the efforts that the authors have taken in this revision, and the very detailed response to reviewers.

I should first state that I recognize the complexity of the dataset, the noisy nature of ecological data, and the constraints of limited samples. Given that it is novel to ask these questions about BEF relationships for a chemosynthetic ecosystem, this paper is a valuable contribution regardless of sample size. That said, my feeling is that the authors are taking a statistical sledgehammer to a conceptually straightforward question.

Bottom line, with only 39 samples, I don't think that a model with more than 10 initial predictors (some of which are being modeled using 3 separate smoothers) can be justified.

I recommend that the authors step back and consider their core question – how does diversity influence ecosystem function, and avoid trying to control for every possible variable. The authors lay out a very clear hypothesis in Figure 1, and then go about testing that hypothesis in a very convoluted manner.

From this perspective, the authors have 3 measures of species diversity and 3 measures of functional diversity (or if they choose the reduced variables from the PCA), and two measures of productivity (standing biomass and activity classification). Simple gam models between these variables can be justified a-priori and are likely all that the authors need to answer their core question. This will most likely bring the authors back to the place that they get to in Figure 2 without a whole lot of statistically questionable intermediaries.

A separation of the Species diversity and functional diversity variables may be valuable. Yes, they may be highly correlated, but conceptually they measure very different things. Modeling conceptually different variables separately may help to produce clearer results.

Thank you very much for this statistical advice. Upon your suggestion, we have indeed taken a step back and re-examined the models. Our changes are also documented in detail below, but in summary we:

- Agree that modelling taxonomic and functional trait diversity separately may help to produce clearer results, and so have separated out these aspects of biodiversity and modelled them independently.
- Investigated redefining some variables as 'random'. Upon this, it became clear that the model structure could be simplified *a priori* since some variables added complexity, but very little further information in reality. For example, 'Cruise ID' and 'Methane Seep Site' were included in original models to control for variability between sampling years, and variability amongst seeps. However, this information is also captured by the oceanographic variables investigated (depth, temperature, salinity, oxygen concentration, export of organic carbon to seep depth). We therefore deemed it more appropriate to remove such variables (cruise identity and methane seep site) from the initial models, removing the need to model them as random effects, and simplifying our statistical approach.

Minor comments:

I appreciate the provision of the R scripts. I had numerous questions about the analytical approach that were unclear in the text but clear in the scripts.

L258-262. It would be very helpful in this paragraph to provide an overview of the analytical approach, particularly to indicate the response and explanatory variable(s) for the gams. As it stands, the following paragraphs require a lot of back and forth because it is not immediately clear what the variables being generated by the PCAs are being used for. I recognize that many of these details are included in the supplementary information, but it is important to have a broad overview of the analytical approach in the main text.

This paragraph (lines 234 - 251) has been revised to clarify the structure of the models analysed.

Dependent and independent variables are now clearly marked:

“The form of relationship *between biodiversity variables and ecosystem functioning proxies* at active chemosynthetically-dependent, transitional ‘chemotone’, and background photosynthetically-dependent habitats was investigated via a Generalised Additive Model (GAM) framework in R v3.4.2 using the package ‘mgcv 1.8-33’ (47, 51). *Please note that GAMs reveal correlation only, and thus no conclusions can be made regarding causal relationships between the variables investigated. Here, ecosystem functioning proxies were modelled as the dependent variable, whilst biodiversity variables, and variables documenting facets of the environment, were modelled as independent variables.* Prior to analysis, continuous variables were standardised, missing environmental data (oceanographic variables and terrain variables (Table 1); 4 values of 304 observations) were replaced with ‘0’ after standardisation, and dimensionality in the dataset was reduced by undertaking Principal Component Analysis (PCA) in R v3.4.2. For ecosystem functioning proxies (*dependent variable*), the first two principal components captured 97.4 % of variance, whilst for *taxonomic biodiversity and functional biodiversity (independent variables)*, the first two principal components captured 98.4 % and 100.0 % of variance, respectively (Table 1, Table 2). For oceanographic variables (*independent variable*), the first two principal components captured 99.5 % of variance, whilst for terrain variables (*independent variable*), the first principal component captured 99.5 % of variance (Table 1, Table 2). Please see Table S5 for details of variable weighting on each PC axis.

Smoothers were specified for all continuous independent variables (Table S6), and optimised automatically using the Generalised Cross Validation criterion (51).”

Please may I also draw the reviewer’s attention to Table S6, which now clearly details model structure.

L277 Again this is confusing because it has not yet been explicitly specified what the explanatory variable(s) in the gams are.

Please see above – we have now clarified dependent and independent variables in the manuscript text.

Supplement L116 and on. It appears that there are a number of variables that could well be considered random terms here (e.g. cruise). It is not clear to me why these variables even need to be in these models, and if they are included why, they are not included as a random term.

We initially investigated and constructed Generalised Additive Mixed Models, with ‘methane seep site’ and ‘cruise ID’ modelled as random terms. However, when examining model fit and refining model structure, it became clear to us that these variables added unnecessary complexity to the models; the information they impart (relating to seasonal differences and oceanographic differences amongst sites) is also contributed by the oceanographic variables investigated (depth, temperature, salinity, oxygen concentration, export of organic carbon to seep depth). As suggested, we therefore elected to construct simplified GAMs based on a reduced number of independent variables *a priori*. Specifically, ‘methane seep site’ and ‘cruise ID’, along with delta oxidation/ reduction potential, have been removed from starting models as they added unnecessary complexity to models without imparting important information.

Referee: 4

This is my second review of this manuscript; my first was as a specialist reviewer to assess the analytical and statistical approaches. I mostly limited my previous comments to the SEM model that was being used to analyze the data, which I found inappropriate based on the small sample size and assumptions of the model. I appreciate that the authors took my comments and suggestions seriously and selected an alternative approach for their analysis. I, like at least one other reviewer of this paper, am a terrestrial plant and animal community ecologist with a history of studying diversity-productivity and diversity-ecosystem function relationships. Consequently, I cannot comment on the specific findings or novelty of the paper that relate to benthic marine communities or chemosynthetic habitats, but I can comment on the aspects of the paper that relate to the theory and methods of biodiversity-ecosystem function research.

Major comments:

The strength of this manuscript lies in the sampling of rare and novel habitats which create a strong gradient in energy input and, qualitatively, in the source of the energy input (light versus chemosynthetic). Even though I don't know this ecosystem or the deep-sea literature, I do feel like the gradient of habitats sampled and the data collected represent a potentially important contribution to the and improve our understanding of how this unique ecosystem works. However, I feel that casting this as a paper that can teach us something general and broad about the biodiversity - ecosystem function (BEF) relationship is misguided. The small sample size, lack of experimental manipulations, static nature of the data collection (i.e., no time-series data), and lack of true measures of ecosystem services make for a poor fit to answer BEF types of questions. There have literally been hundreds of BEF papers and some of them have used high dimensional models to analyze thousands of samples from experiments distributed across the globe. So, while I think the gradient of energy or resources creates an interesting template across which to explore patterns of richness, density, biomass and species traits I think the results and conclusions go too far in attempting to shed light on general BEF theory in what amounts to a very unique habitat.

We greatly appreciate this alternative perspective on our study, and agree that the manuscript (in its past form), did not fully play to its strengths of strong natural gradient in energy, alternative source of energy input (chemosynthesis vs light), and little-studied habitat. We have therefore made a series of edits to the manuscript, reframing it as a mensurative study, reducing discussion of what our findings might tell us about biodiversity – ecosystem functioning relationships in a broad sense, and tightening focus around the natural gradient in energy and unusual habitat that the study investigated. Specifically,

Abstract: We emphasised that the main novelty in this manuscript lies in the sampling of rare and little-studied deep-sea chemosynthetic methane seep habitats, and the investigation of the effect of the strong gradient in resource supply surrounding them on biological communities: “Here, we utilise deep-sea chemosynthetic methane seeps and surrounding sediments as natural laboratories in which to contrast relationships between biodiversity and ecosystem functioning proxies along a gradient of resource availability (higher-resource – organic carbon – methane seep, to lower-resource photosynthetically-fuelled deep-sea habitats)”. We clarified that our results shed light on the ecology of deep-sea methane seep environments, but should not be extrapolated to other habitats more generally: “This suggests that *absolute biodiversity is not a good metric of ecosystem ‘value’ at methane seeps, and that these deep-sea environments may require special management to maintain ecosystem functioning under human disturbance...*[we] emphasise that *deep-sea conservation efforts should consider ‘functioning hotspots’ alongside biodiversity hotspots*”.

Introduction: We have emphasised the novel environment in which this study takes place e.g. lines 71-73 “However, very few of these studies have investigated relationships between biodiversity and ecosystem functioning proxies in the deep ocean, and no studies have investigated these relationships

in chemosynthetic environments". Lines 117-126 "Here, we undertake a mensurative study, utilising deep-sea methane seeps and the surrounding seafloor as a natural laboratory *in which* to investigate how the form of relationship *between biodiversity and ecosystem functioning proxies* changes along gradients of resource supply; from lower-resource deep-sea habitats primarily dependent upon the delivery of photosynthetically-fixed carbon from surface waters, to higher-resource chemosynthetically-fuelled seep habitats. Deep-sea methane seeps *are resource* hotspots in what is typically considered a food-limited environment (41-43). Steep gradients of resource supply and origin surround methane seeps, as well as gradients in other environmental variables, including sediment and water chemistry (and hence physiological stress associated with reducing fluids low in oxygen and high in hydrogen sulphide concentrations), substratum type, and turbidity (41)".

We have removed discussion of causal relationships between biodiversity loss and loss of ecosystem functioning. For example, the following text has been removed: "For example, in the case of saturating BEF relationships, a high degree of biodiversity loss can be sustained before ecosystem functioning declines substantially (Figure S1; Supplementary Materials). In contrast, for accelerating BEF relationships (e.g. exponential relationships) even a small loss of biodiversity may result in a significant decline in ecosystem functioning (Figure 1, Figure S1)".

As covered in full detail in a comment below, we have further revised all wording through the manuscript that may suggest causality in the relationships investigated, including the title of the manuscript, and have clarified that our hypotheses relate to bivariate relationships between biodiversity and proxies for ecosystem functioning, and do not make claims of causality in these relationships: "We hypothesised that the *shape* of relationship *between biodiversity and ecosystem functioning proxies* would change when moving from photosynthetically-dependent deep-sea habitats towards chemosynthetically-dependent methane seep habitats (Figure 1). We predicted that relationships between *biodiversity and ecosystem functioning proxies* at methane seeps (high resource supply) would be positive and saturating *in shape*, whilst *relationships between biodiversity and ecosystem functioning proxies* in background habitats (low resource supply) would be positive and accelerating, and relationships *between biodiversity and ecosystem functioning proxies* in the transitional 'chemotone' between these habitats (moderate resource supply (44)) would be linear *in shape* (20) (Figure 1)".

We have removed wording that may give readers a false sense of how our results may be generalised. For example, the following wording has been removed from the end of the introduction: "Our findings promote the value of investigating BEF relationships under a range of resource sources in varied environments, and lead us to emphasise that conservation efforts and management practices should consider 'functioning hotspots' alongside the biodiversity hotspots that are more commonly targeted". Instead, we have inserted the following sentence at line 419 of the Discussion: "*Future studies should investigate the relationship between biodiversity and ecosystem functioning at other marine chemosynthetic environments, such as hydrothermal vents*".

Methods and results: We have emphasised throughout the methods and results that our mensurative study investigates only correlations between biodiversity and proxies for ecosystem functioning (please see response below for further discussion of these proxies) – not causal relationships between biodiversity and the provision of ecosystem services. The variables used as proxies for ecosystem functioning are only ever referred to as proxies, never as direct measures of ecosystem functioning or ecosystem services.

Discussion: We have revised wording that may mislead readers into thinking that we have investigated causal relationships between biodiversity and ecosystem functioning. For example, we have changed "Here, we respond to studies calling for further investigation of the influence of environmental conditions over the form of BEF relationships (27, 40, 54). By utilising methane

seepage gradients as a natural laboratory, we explore how the strength and form of relationship between biodiversity and ecosystem functioning proxies is influenced by changing resource characteristics in a little-studied realm that is under increasing threat of disturbance from human resource extraction” to “Here, undertaking a mensurative study, we utilised methane seepage gradients as a natural laboratory in which to explore how the strength and form of relationship between biodiversity and ecosystem functioning proxies is influenced by changing trophic resource characteristics in a little-studied realm that is under increasing threat of disturbance from human resource extraction”. Throughout, we have changed all wording suggesting that we investigated BEF relationships to more accurately state that we investigated correlations between biodiversity and variables that could be considered proxies for ecosystem functioning. For example, we have changed wording at lines 466-468 from “...the loss of a small number of these critical species could result in a major decline in ecosystem functioning” to “...the loss of a small number of these critical species *could be correlated with* a major decline in ecosystem functioning”.

We have revised wording that may mislead readers into generalising our results to other environments and situations. For example, we have revised wording suggesting that our results are directly applicable to other environments, to clarify that they may only be applied with a degree of confidence to methane seep environments (for example, see lines 356, 413, 415, 422, 423, 443, 450, 465). We have removed the following sentence from the conclusion of the manuscript: “Greater understanding of the relationship between biodiversity and ecosystem functioning may be achieved by considering how both the source and magnitude of resource production affects the form of BEF relationships”.

One criticism which I think needs to be addressed is how this manuscript treats and defines ecosystem services in the first place. In all ecosystem function literature which I have read (which amounts to quite a lot over the years) the quantifiable “services” are typically processes that are measured by rates. Typical measures include variables like NPP, rates of biogeochemical cycling, flood control, etc., all which can be measured as an amount of something that is produced, cycled or mitigated over time. But here, the authors use at least three ecosystem functions that to me are perplexing: faunal density, faunal biomass and individual body size. The other two metrics, which include bioturbation potential and calcification metric, appear to be much more in line with what I expect are typical ecosystem services. But the other three are very problematic because, while they may be correlated with rates of energy flow or biochemical transformations (although not necessarily), they are not surrogates for ecosystem services and in fact they are confounded in important ways.

Body size should be thought of as a species trait that is shaped by evolution and life history. Indeed, your citation list supports that this is a species trait rather than an ecosystem services. For example, the paper by Chapman et al. “sFDvent: A global trait database for deep-sea hydrothermal-vent fauna” listed body size as a key species trait that can influence ecosystem function, not a measure of ecosystem function itself.

With respect to faunal density, I attempted to raise this issue in my previous review but perhaps it was missed because of the many other comments: the cause and effect of richness and faunal density appear to be confounded in this manuscript. Faunal density is being measured as an ecosystem service (i.e., as a response variable) that is causally related to species richness and diversity, whereas the reverse causal relationship is equally likely to be true. In plant communities it has often been observed that as the number of individuals in a patch increases so too does the number of species because of the sampling effect. At high density strong competitive effects then reduce the number of species in a patch. So the question becomes what biotic or abiotic factors are controlling the density of species in communities, an issue raised by the authors themselves in lines 83 – 88. Ecologists have long discussed these issues and several well-established methods, such as the well-known 2001 paper by Gotelli and Colwell in *Ecology Letters*, which describes sample- and individual-based rarefaction and

accumulation curves to compare richness while correcting for differences in density. But even more problematic for the current manuscript is the fact that the authors acknowledge that energy or resource inputs should control species density but then they use species density as an ecosystem service (i.e. a response variable) that putatively responds to diversity. The result is that the variables are highly confounded, begging the question of: what variable is causally controlling what other variables?

With respect to biomass, the problem is similar to density: what is the factor that ultimately controls the biomass, is it resources or diversity? As is rightfully acknowledged in the introduction, including lines 96-101, resources influence biomass accumulation which controls species richness. In the BEF literature that is cited throughout the manuscript (i.e., Tilman, Naeem, Hector, Hooper, etc.) the central question was, after controlling for the negative effect of biomass on diversity, how much of a residual effect of diversity is there on the biomass and primary production? The highly correlated and confounded nature of density, diversity and function are what drove ecologists to tease apart these relationships with well-planned highly controlled experiments such as the European BIODEPTH experiment, Cedar Creek LTER and nutnet.

I don't doubt the bivariate shape of the relationships described in the results section of the manuscript, but I do doubt that one can assign any causal explanation for what gives rise to the shape of these relationships without more data and a better model.

So, in the end my main suggestion is to reframe the paper as an observational exploration of how these bivariate relationships change across gradients of energy and resource inputs and tone down the discussion of BEF, which as I have expressed, I think is poorly addressed by the data and experimental design. Alternatively one could focus truly on services that are not confounded with density and maintain the hypotheses based on BEF. The authors are in a better position to know if bioturbation and biogenic calcium carbonate content are clear representations of ecosystem services, but density and body size must be eliminated as focal ecosystem services. More appropriate is to analyze density and body size as stand-alone response variables (which then answers a very different ecological question) or as covariates in an analysis focused on other response variables.

Thank you very much for highlighting these important points. In response, we would like to emphasise that the study does not attempt to directly quantify ecosystem services. Instead, we quantify only proxies for ecosystem functions. These proxies may ultimately be linked to ecosystem services, or may not. To avoid misunderstanding, we are defining ecosystem functioning here as 'the aggregate effect of physical, chemical and biological processes associated with the flow of energy and matter within and between trophic levels, communities and ecosystems', and ecosystem services as 'the benefits that people obtain from ecosystems'. To clarify the relationships amongst biodiversity, ecosystem functioning and ecosystem services in the manuscript so that readers fully recognise that the variables quantified in the study represent proxies for functions only (not an attempt to directly quantify ecosystem services) we have changed lines 55-57 to "The ecosystem services *associated with* biologically diverse, *efficiently functioning* ecosystems are numerous, economically valuable, and fundamental to human wellbeing worldwide (1-4)". We have removed reference to ecosystem services at lines 65, 347 and 456 to avoid giving readers a false sense that our study results can be extrapolated to the provision of ecosystem services. We have revised Table S3 to include more detailed information regarding the relevance of variables as proxies for ecosystem functioning. We have critically reviewed all usage of the term 'ecosystem functioning' in the manuscript, adding emphasis that the variables analysed can only be considered as simple proxies for ecosystem functioning at lines 182, 234, 239, 291, 293, 295, 297- 312, 315, 327, 333, 336, 340-341, 358, 358, 371, 373, 400, 413, 416, 434, 440, 472, and in Figure 2, Table 1, Table 2 and Table 3.

We agree that original use of body size as a proxy for ecosystem functioning was tenuous, and that it is more appropriate to consider body size as a trait. We have therefore completely removed this variable from our selection of ecosystem functioning proxies. Note that body size information is used in the study as a component in the calculation of functional trait diversity (please see Dataset S1).

We agree that the direction of causality is complex in the relationships investigated. Considering this alongside the fact that the new analysis approach of using GAMs does not assume or give indication of causality in relationships, we have reviewed all wording in the manuscript to remove any suggestion of causality in the relationships investigated. Specifically, we have changed the title of the manuscript to “*Relationships between biodiversity and ecosystem functioning proxies strengthen when approaching chemosynthetic deep-sea methane seeps*”. We have removed reference to biodiversity ‘driving’ ecosystem functioning in methane seeps at line 45 (abstract) and lines 369, 375, 387, 402 and 466 (discussion). We have clarified from lines 117 – 122, and lines 128 - 136 that this manuscript investigates relationships between biodiversity and proxies for ecosystem functioning in deep-sea environments, as opposed to investigating causal BEF relationships. For example, lines 128 - 136 now read “We hypothesised that the *shape of relationship between biodiversity and ecosystem functioning proxies* would change when moving from photosynthetically-dependent deep-sea habitats towards chemosynthetically-dependent methane seep habitats (Figure 1). We predicted that *relationships between biodiversity and ecosystem functioning proxies* at methane seeps (high resource supply) would be positive and saturating *in shape*, whilst *relationships between biodiversity and ecosystem functioning proxies* in background habitats (low resource supply) would be positive and accelerating, and *relationships between biodiversity and ecosystem functioning proxies* in the transitional ‘chemotone’ between these habitats (moderate resource supply (44)) would be linear *in shape* (20) (Figure 1)”. Similarly, we have clarified at lines 357 – 359 that the manuscript investigates the strength and shape of correlation between biodiversity and ecosystem functioning proxies, as opposed to investigating causal BEF relationships. Throughout the discussion section, instead of referring to our results as ‘BEF relationships, we have now clarified that the models investigated the strength and shape of relationship between biodiversity and proxies for ecosystem functioning (for example, see lines 358, 361, 385, 393, 400, 413, 416, 434, and 472). Reducing reference to causality, lines 441- 444 now read “This variation in form of relationship [between biodiversity and ecosystem functioning proxies] with changing habitat suggests that biodiversity loss in contrasting deep-sea environments *may be correlated with differing changes in ecosystem functioning*”.

With regards to faunal density and standing stock, we agree that these factors will tend to increase in magnitude with increasing energy input. However, further to this, we are interested in investigating whether variation in species or trait presence/relative abundance at a set energy input will also alter faunal density and biomass. An example, consider a microcosm where energy resources are tightly controlled to a set level; will increasing microbial diversity impact microbial density and standing stock? Deep water studies have shown that facilitation is an important force in determining community structure (for example, please see Danovaro *et al.* 2008 ‘Exponential decline of deep-sea ecosystem functioning linked to benthic biodiversity loss’ *Current Biology*; Levin *et al.* 1997 ‘Rapid subduction of organic matter by maldivian polychaetes on the North Carolina slope’ *Journal of Marine Research*; Govenar *et al.* 2004 ‘Composition of a one-year-old *Riftia pachyptila* community following a clearance experiment: Insight to succession patterns at deep-sea hydrothermal vents’ *The Biological Bulletin*; and Deng *et al.* 2020 ‘Macrofaunal control of microbial community structure in continental margin sediments’ *PNAS*). This helped form our hypotheses regarding relationships between biodiversity faunal density/ standing stock. We agree that understanding causality here is very difficult, but believe that it is legitimate to examine these relationships as part of the models constructed in this study. In recognition of the complexity of assigning causality to these relationships, as detailed above, we have removed any wording in the manuscript which might mislead readers into assuming causal relationships between biodiversity and the ecosystem functioning proxies investigated. To further emphasise this, we have also added the following text to

the methods section (lines 237 - 238) “Please note that GAMs reveal correlation only, and thus no conclusions can be made regarding causal relationships between the variables investigated”.

We believe that our reframing of the manuscript as a mensurative study, greater focus on the unique study system investigated, and reduced extrapolation of findings to alternative environments together address concerns regarding assigning undue causality to the relationships investigated, and ‘overstretching’ of conclusions.

If the authors choose to continue to frame their hypotheses in terms of ecosystem function, there are some potentially important missing references from this paper that may help to clarify their questions. These include the 1994 “green book” edited by ED Schulze and HA Mooney entitled “Biodiversity and Ecosystem Function” (and references therein) and a 1996 book (and references therein) edited by Mooney, Cushman, Medina, Sala and Schulze titled “Functional roles of biodiversity: a global perspective.” The idea that ecosystem function is influenced by species loss or gain, and that the nature of this relationship is influenced by which species are lost from or added to a system, was described by Vitousek and Hooper in the green book and in a model by Sala et al. in the 1996 Mooney et al. book. It is my feeling that the authors of this paper need to read some of this classic BEF literature to provide appropriate context for their ideas and hypotheses. I know that their newer citations are based on these more classic texts, but I was surprised none showed up in the reference list.

We thank the reviewer for this suggestion. We agree that these texts deserve reference at multiple points in the manuscript, and have cited them at lines 65 – 67, 90, and 92.

Minor comments:

Line 79-81 and Figure 1: see my note above about missing literature, references and ideas previously described in the BEF literature.

Thank you. As mentioned above, we have now given the origins of these concepts due credit in the manuscript.

Line 72-73: This seems to be a reference about deep sea ecosystem. If so, please clarify and add more detailed information about the findings of citation 19.

Citation 19 references the lead author’s PhD thesis, a chapter of which (entitled “What is the form of deep-sea biodiversity – ecosystem functioning relationships”) investigated relationships between crustacean biodiversity metrics and proxies for ecosystem function in the deep waters of the NW Atlantic. The chapter concluded that the shape of relationship between biodiversity and ecosystem functioning was likely context-dependent. The thesis is available here: <https://ora.ox.ac.uk/objects/uuid:228c4d19-56a8-41e1-a1da-9ca13fe2eef1>. The deep-sea nature of this reference has been expanded upon at lines 103 - 108: “In deep-sea environments, this has been hypothesised to produce BEF relationships which saturate quickly with increasing diversity (20). In contrast, when resource supply is relatively low, organismal densities and assemblage dominance are expected to be lower, promoting positive species interactions, such as facilitation, and high assemblage diversity. In deep-sea environments, this has been hypothesised to produce BEF relationships which become increasingly positive with increasing diversity (18, 20).

Line 90: I don’t agree that resource and energy gradients are synonymous as assumed here, but it would be interesting to hear why the authors thought this was so. This line deserves a bit more information and perhaps some citations. I recommend a book edited by RE Ricklefs and D Schluter titled “Species Diversity in Ecological Communities” and (references therein) as well as richness-energy papers published by David Currie in Am Nat. To me energy and resources differ based on scale. Energy enters ecosystems via production but it is not something species compete for per se.

Resources on the other hand are being competed for and display heterogeneity over smaller scales than energy.

The insertion of ‘energy’ at line 90 was the suggestion of another reviewer in the prior iteration of the manuscript, who stated “*As a terrestrial plant ecologist, I find figure 4 curious because I normally think of productivity as an ecosystem function. I wonder if characterizing these as high energy input and low energy input systems would be a better approach*”.

Upon further consideration based on your comments above, we agree that the gradient examined would be better described just as a trophic resource gradient, and have revised the manuscript at lines 36, 94, 352 to reflect this.

Lines 246-249: You mentioned primary productivity of the sites and listed Table 1 and Table S4 but I never again saw mention of this important variable and I did not see it presented in either of the tables. I also failed to find it in dataset S3. This seems like an important covariate or predictor that should be part of the analysis but I think it was omitted.

Lines 246-249 of the manuscript reviewed reference our estimation of surface water primary production that was exported to the location and depth of each sampling point. This variable is encompassed in the ‘oceanography’ PCA, and was present at line 11 of Table 1, line 14 of Table S4, and as variable ‘Seafloor_POC_delivery_mg/m³/month’ in dataset S3. We agree that inconsistency in the naming of this variable is confusing, and so have revised the manuscript and supplementary materials so that it is referred to as ‘export of surface productivity to seafloor’ throughout (or a shortened version for ‘dataset S3’).

Lines 298-317: Reporting the correlations with PCA factors is important, but I found this section lengthy and not to be composed of particularly interesting ecological results. This is almost more of a technical result that could be greatly shortened and largely presented in the supplemental materials. As I discussed at length above, the creation of PCA variables that are of such differing and contrasting nature is problematic and should be reconsidered.

We agree with this assessment, so have moved this text to supplementary materials (Section 1.4), and have presented the raw weightings of variables on PC axes as Table S5. The reader is now referred to this supplementary information in the main manuscript at line 250.

Please see comments above for changes we have made to the models analysed, and also note (as detailed in our response to the comments of the other reviewer), we have now split functional trait and taxonomic metrics of biodiversity into separate models.